# Dual Generator Offline Reinforcement Learning

**Quan Vuong**[1]**, Aviral Kumar**[1,2]**, Sergey Levine**[1,2]**, Yevgen Chebotar**[1]

[1]Google Research, [2]UC Berkeley

## Abstract

In offline RL, constraining the learned policy to remain close to the data is essential to prevent the policy from outputting out-of-distribution (OOD) actions with erroneously overestimated values. In principle, generative adversarial networks (GAN) can provide an elegant solution to do so, with the discriminator directly providing a probability that quantifies distributional shift. However, in practice, GAN-based offline RL methods have not performed as well as alternative approaches, perhaps because the generator is trained to both fool the discriminator and maximize return – two objectives that can be at odds with each other. In this paper, we show that the issue of conflicting objectives can be resolved by training two generators: one that maximizes return, with the other capturing the "remainder" of the data distribution in the offline dataset, such that the mixture of the two is close to the behavior policy. We show that not only does having two generators enable an effective GAN-based offline RL method, but also approximates a support constraint, where the policy does not need to match the entire data distribution, but only the slice of the data that leads to high long term performance. We name our method DASCO, for **D**ual-Generator **A**dversarial **S**upport **C**onstrained **O**ffline RL. On benchmark tasks that require learning from sub-optimal data, DASCO significantly outperforms prior methods that enforce distribution constraint.

## 1 Introduction

Offline reinforcement learning (RL) algorithms aim to extract policies from datasets of previously logged experience. The promise of offline RL is to extract *decision making engines* from existing data [32]. Such promise is especially appealing in domains where data collection is expensive or dangerous, but large amounts of data may already exists (e.g., robotics, autonomous driving, task-oriented dialog systems). Real-world datasets often consist of both expert and sub-optimal behaviors for the task of interest and also include potentially unrelated behavior corresponding to other tasks. While not all behaviors in the dataset are relevant for solving the task of interest, even sub-optimal trajectories can provide an RL algorithm with some useful information. In principle, if offline RL algorithms can combine segments of useful behavior spread across multiple sub-optimal trajectories together, the combined segments can then perform better than any behavior observed in the dataset.

Effective offline RL requires estimating the value of actions other than those that were taken in the dataset, so as to pick actions that are better than the actions selected by the behavior policy. However, this requirement introduces a fundamental tension: the offline RL method must generalize to new actions, but it should not attempt to use actions in the Bellman backup for which the value simply cannot be estimated using the provided data. These are often referred to in the literature as out-of-distribution (OOD) actions [28]. While a wide variety of methods have been proposed to constrain offline RL to avoid OOD actions [26, 14, 1], the formulation and enforcement of such constraints can be challenging, and might introduce considerable complexity, such as the need to explicitly estimate the behavior policy [49] or evaluate high-dimensional integrals [30]. Generative adversarial networks (GANs) in principle offer an appealing and simple solution: use the discriminator to estimate whether an action is in-distribution, and train the policy as the "generator" in the GAN formulation to fool

this discriminator. Although some prior works have proposed variants on this approach [49], it has been proven difficult in practice as GANs can already suffer from instability when the discriminator is too powerful. Forcing the generator (i.e., the policy) to simultaneously *both* maximize reward and fool the discriminator only exacerbates the issue of an overpowered discriminator.

We propose a novel solution that enables the effective use of GANs in offline RL, in the process not only mitigating the above challenge but also providing a more appealing form of support constraint that leads to improved performance. Our key observation is that the generative distribution in GANs can be split into *two* separate distributions, one that represents the "good parts" of the data distribution and becomes the final learned policy, and an auxiliary generator that becomes the policy's complement, such that their mixture is equal to the data distribution. This formulation removes the tension between maximizing rewards and matching the data distribution perfectly: as long as the learned policy is within the *support* of the data distribution, the complement will pick up the slack and model the "remainder" of the distribution, allowing the two generators together to perfectly fool the discriminator. If however the policy ventures outside of the data support, the second generator cannot compensate for this mistake, and the discriminator will push the policy back inside the support. We name our method DASCO, for **D**ual-Generator **A**dversarial **S**upport **C**onstrained **O**ffline RL.

Experimentally, we demonstrate the benefits of our approach, DASCO, on standard benchmark tasks. For offline datasets that consist of a combination of expert, sub-optimal and noisy data, our method outperforms distribution-constrained offline RL methods by a large margin.

## 2 Related Work

Combining sub-optimal trajectories to obtain high-performing policies is a central promise of offline RL. During offline training, querying the value function on unseen actions often leads to value over-estimation and collapse in learning progress. To avoid querying the value functions on out-of-distribution actions, existing methods encourage the learned policies to match the distribution of the behavior policies. This principle has been realized with a variety of practical algorithms [22, 49, 41, 42, 49, 29, 26, 25, 47, 12, 9, 15, 21, 37, 10, 34]. For example, by optimizing the policies with respect to a conservative lower bound of the value function estimate [30], only optimizing the policies on actions contained in the dataset [26], or jointly optimizing the policy on the long-term return and a behavior cloning objective [12]. While *explicitly* enforcing distribution constraint by adding the behavior cloning objective allows for good performance on near-optimal data, this approach fails to produce good trajectories on sub-optimal datasets [26]. Methods that *implicitly* enforce distribution constraints, such as CQL and IQL, have seen more successes on such datasets. However, they still struggle to produce near-optimal trajectories when the actions of the dataset generation policies are corrupted with noise or systematic biases (a result we demonstrate in Section 5).

However, enforcing distribution constraints to avoid value over-estimation may not be necessary. It is sufficient to ensure the learned policies do not produce actions that are too unlikely under the dataset generation policy. That is, it is not necessary for the learned policy to fully *cover* the data distribution, only to remain in-support [29, 27, 32, 49, 51, 8]. Unfortunately, previous methods that attempt to instantiate this principle into algorithms have not seen as much empirical success as algorithms that penalize the policies for not matching the action distribution of the behavior policies. In this paper, we propose a new GAN-based offline RL algorithm whose use of dual generators naturally induce support constraint and has competitive performance with recent offline RL methods. In a number of prior works, GANs have been used in the context of imitation learning to learn from expert data [20, 33, 19, 35]. In this work, we show that dual-generator GANs can be used to learn from sub-optimal data in the context of offline RL.

To enforce the support constraint, we use the discriminator score to measure how likely an action is under the behavior policy. To measure uncertainty, one could also use an ensemble of value functions, as is done in [2, 3]. We include [2] as a baseline in Table 1 and demonstrate the our method has significantly higher performance. Nevertheless, it would be interesting as future work to combine the discriminator score and an ensemble of value functions to devise a higher performing algorithm.

## 3 Background

Let $\mathcal{M} = (\mathcal{S}, \mathcal{A}, P, R, \gamma)$ define a Markov decision process (MDP), where $\mathcal{S}$ and $\mathcal{A}$ are state and action spaces, $P : \mathcal{S} \times \mathcal{A} \times \mathcal{S} \to \mathbb{R}_+$ is a state-transition probability function, $R : \mathcal{S} \times \mathcal{A} \to \mathbb{R}$ is a

reward function and $\gamma$ is a discount factor. Reinforcement learning methods aim at finding a policy $\pi(a|s)$ that maximizes the expected discounted reward $R(\tau) = \sum_{t=0}^{T} \gamma^t R(s_t, a_t)$ over trajectories $\tau = (s_0, a_0, \ldots, s_T, a_T)$ with time horizon $T$ induced by the policy $\pi$.

In this work, we concentrate on the offline or off-policy RL setting, i.e. finding an optimal policy given a dataset $\mathcal{D}$ of previously collected experience $\tau \sim \mathcal{D}$ by a behavior policy $\pi_\beta$. A particularly popular family of methods for offline learning are based on training a Q-function through dynamic programming using temporal-difference (TD) learning [48, 44]. Such methods train a Q-function to satisfy the Bellman equation:

$$Q(s_t, a_t) = R(s_t, a_t) + \gamma \mathbb{E}_{a \sim \pi}[Q(s_{t+1}, a)].$$

In Q-learning, the policy is replaced with a maximization, such that $\pi(a|s) = \arg\max_a Q_\theta(s, a)$, while actor-critic methods optimize a separate parametric policy $\pi_\phi(a|s)$ that maximizes the Q-function. In this work, we extend the Soft Actor-Critic (SAC) method [18] for learning from diverse offline datasets.

Generative Adversarial Networks (GANs) [17] enable modeling a data distribution $p_\mathcal{D}$ through an adversarial game between a generator $G$ and a discriminator $D$:

$$\min_G \max_D \mathbb{E}_{x \sim p_\mathcal{D}}[\log(D(x))] + \mathbb{E}_{z \sim p(z)}[\log(1 - D(G(z)))] \tag{1}$$

For this two player zero-sum game, [17] shows that for a fixed generator $G$, the optimal discriminator is $D_G^*(x) = \dfrac{p_\mathcal{D}(x)}{p_\mathcal{D}(x) + p_G(x)}$ and the optimal generator matches the data distribution $p_g^*(x) = p_\mathcal{D}$.

GAN has been extended to the offline RL setting by interpreting the discriminator function as a measure of how likely an action is under the behavior policy, and jointly optimizing the policy to maximize an estimate of the long-term return and the discriminator function [49]:

$$\min_\pi \max_D \mathbb{E}_{s,a \sim p_\mathcal{D}}[\log(D(s, a))] + \mathbb{E}_{s \sim p_\mathcal{D}, a \sim \pi(a|s)}[\log(1 - D(s, a))] - \mathbb{E}_{s \sim p_\mathcal{D}, a \sim \pi(a|s)}[Q(s, a)], \tag{2}$$

where $Q(s, a)$ is trained via the Bellman operator to approximate the value function of the policy $\pi(a|s)$. This leads to iterative policy evaluation and policy improvement rules for the actor and the policy [49]. During the $k^{th}$ update step, given the most recent values for the policy $\pi^k$, the value function $Q^k$, and the discriminator $D^k$, we perform the following updates to obtain the next values for the value function and the policy:

$$
\begin{aligned}
Q^{k+1} &\leftarrow \arg\min_Q \mathbb{E}_{s,a,s' \sim \mathcal{D}} \left[ \left( (R(s, a) + \gamma \mathbb{E}_{a' \sim \pi^k(a'|s')}[Q^k(s', a')]) - Q_{target}(s, a) \right)^2 \right] \\
\pi^{k+1} &\leftarrow \arg\max_\pi \mathbb{E}_{s \sim \mathcal{D}, a \sim \pi^k(a|s)} \left[ Q^{k+1}(s, a) + \log D^k(s, a) \right]
\end{aligned}
\tag{3}
$$

where the $\log D(a|s)$ term in the policy objective aims at regularizing the learnt policy to prevent it from outputting OOD actions. In practice, training the policy to maximize both the value function and discriminator might lead to conflicting objectives for the policy and thus poor performance on either objective. This can happen when the data contains a mixture of good and bad actions. Maximizing the value function would mean avoiding low-reward behaviors. On the other hand, maximizing the discriminator would require outputting all in-distribution actions, including sub-optimal ones. Our approach alleviates this conflict and enables *in support* maximization of the value function when learning from mixed-quality datasets.

# 4   Dual-Generator Adversarial Support Constraint Offline RL

We now present our algorithm, which uses a novel dual-generator GAN in combination with a weighting method to enable GAN-based offline RL that constrains the learned policy to remain within the support of the data distribution. We call our method *Dual-generator Adversarial Support Constraint Offline RL (DASCO)*. We will first introduce the dual-generator training method generically, for arbitrary generators that must optimize a user-specified function $f(x)$ within the support of the data distribution in Section 4.1. We will then show this method can be incorporated into a complete offline RL algorithm in Section 4.2 in combination with our proposed weighting scheme, and then summarize the full resulting actor-critic method in Section 4.3.

## 4.1 Dual generator in-support optimization

In this section, we will develop an approach for performing a joint optimization of adversarial and secondary objectives of the generator in a GAN framework, which we will then apply to offline RL. This is a necessary component for performing the joint optimization in Eq. 2 without introducing a conflict of these objectives. All proofs for theorems presented in this section are in Appendix A.

Let's consider a general objective that requires training a generator $G$ to fool the discriminator $D$ while also optimizing the expected value of some other function $f$:

$$\min_G \max_D \quad \mathbb{E}_{x \sim p_{\mathcal{D}}}[\log(D(x))]$$
$$+ \mathbb{E}_{z \sim p(z)}[\log(1 - D(G(z)))]$$
$$+ \mathbb{E}_{z \sim p(z)}[f(G(z))] \tag{4}$$

where the first two terms are the same as the objective of the GAN formulation. We have also added an additional term $\mathbb{E}_{z \sim p(z)}[f(G(z))]$, where $f$ is a mapping from the generator output to a scalar value. The third term represents a secondary objective that the generator should optimize.

**Theorem 4.1** *The optimal generator of Eq. 4 induces a distribution $p_g^*(x) = p_{\mathcal{D}}(x) \dfrac{e^{-f(x)-\nu}}{2 - e^{-f(x)-\nu}}$, where $\nu > 0$ is the Lagrange multiplier that ensures that $p_g^*(x)$ is normalized to 1.*

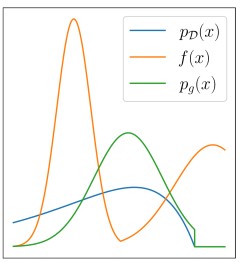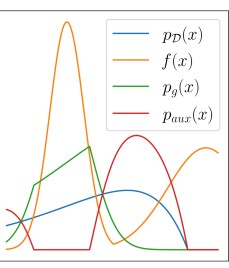

Figure 1: Visualizations to illustrate the benefit of *dual* generators over single generator when maximizing a secondary objective $f(x)$ in the GAN framework. In both figures, $p_{\mathcal{D}}(x)$ is the data distribution. The x-axis is a one-dimensional sample space. **Left:** In this figure, since there is only a single generator, the generator $G$ is trained to jointly maximize the objective $f(x)$ and matches the data distribution $p_{\mathcal{D}}(x)$. The distribution $p_G$ induced by the generator is thus not very good at either maximizing the objective $f(x)$ or matching the data distribution $p_{\mathcal{D}}(x)$. **Right:** In this figure, we have two generators, inducing two distributions $p_G$ and $p_{aux}$. By introducing the auxiliary generator $G_{aux}$ into the GAN framework, the primary generator can better maximize the objective $f(x)$ while staying within the support of the data distribution $p_{\mathcal{D}}$. The mixed distribution also perfectly matches the data distribution, i.e. $\dfrac{p_g(x) + p_{aux}(x)}{2} = p_{\mathcal{D}}(x)$. Note that in these two figures, the primary generator aims to maximize $f(x)$ (instead of minimize) to allow for more intuitive interpretation. In Appendix B, we provide a numerical example to further illustrate the benefit of the dual generator technique on a 1D example.

We can see that by adding a secondary objective function for the generator, in general, the optimal generator does not attempt to match the data distribution $p_{\mathcal{D}}(x)$ anymore, but instead tries to match the data distribution weighted by $\dfrac{e^{-f(x)-\nu}}{2 - e^{-f(x)-\nu}}$. We expect that in such case, the discriminator clearly has an advantage in the two player zero-sum game and will be able to distinguish between real samples and sample generated by the generator.

To allow the generator to specialize in optimizing the secondary objective function, we propose to introduce a second auxiliary generator that matches the portion of the data distribution that is not well captured by the primary generator. Let $p_{mix} = \dfrac{p_g + p_{aux}}{2}$, consider the min-max problem:

$$\min_{G, G_{aux}} \max_D \mathbb{E}_{x \sim p_{\mathcal{D}}}[\log(D(x))] + \mathbb{E}_{x \sim p_{mix}}[\log(1 - D(x))] + \mathbb{E}_{x \sim p_g}[f(x)], \tag{5}$$

where we mix samples from the primary generator $G$ and the auxiliary generator $G_{aux}$ to generate samples that can fool the discriminator. The mixing is indicated by the distribution $p_{mix}$ in the second term of Eq. 5. The first and third term of Eq. 5 are the same as the objective in Eq. 4.

We next theoretically demonstrate the benefit of adding the auxiliary generator to the GAN formulation with the following Theorem.

**Theorem 4.2 (Informal)** *The primary generator $p_G$ performs in-support optimization of $f(x)$.*

We first note that the optimal solution of the mixed distribution from Eq. 5 is the real data distribution:

$$\frac{p_{aux}^*(x) + p_g^*(x)}{2} = p_{\mathcal{D}}(x) \tag{6}$$

Accordingly, the optimal auxiliary generator distribution can be expressed as

$$p_{aux}^*(x) = 2p_{\mathcal{D}}(x) - p_g^*(x) \tag{7}$$

Let $x_0$ to be the element inside the support of the data distribution $p_{\mathcal{D}}$ that minimizes $f$. That is:

$$x_0 = \operatorname*{arg\,min}_{x \in \mathrm{Supp}(p_{\mathcal{D}})} f(x)$$

When optimizing the secondary objective $f(x)$, the primary generator will maximize the probability mass of in-support samples that maximize $f(x)$. However, Eq. 7 introduces a constraint that enforces $2p_{\mathcal{D}}(x) - p_g^*(x) \geq 0$ for $p_{aux}^*(x) \geq 0$ to remain a valid distribution. This leads us to conclude that the optimal primary generator $p_g^*$ assigns the following probability to $x_0$:

$$p_g^*(x_0) = \begin{cases} 2p_{\mathcal{D}}(x_0) & \text{if} \quad 2p_{\mathcal{D}}(x_0) < 1 \\ 1 & \text{otherwise} \end{cases} \tag{8}$$

Interestingly, if the global optimum $x_0$ is not taking the full probability mass, the rest of the probability mass is redistributed to the next best in-support optimum, which we can define recursively:

$$\text{For } x_i \in \operatorname*{arg\,min}_{x \in \mathrm{Supp}(p_{\mathcal{D}}) \setminus \{x_j\}_{j=0}^{i-1}} f(x), \ p_g^*(x_i) = \begin{cases} 2p_{\mathcal{D}}(x_i) & \text{if} \quad \sum_{j=0}^{i} p_g^*(x_j) < 1 \\ 1 - \sum_{j=0}^{i-1} p_g^*(x_j) & \text{if} \quad \sum_{j=0}^{i} p_g^*(x_j) > 1 \\ 0 & \text{if} \quad \sum_{j=0}^{i-1} p_g^*(x_j) = 1 \end{cases} \tag{9}$$

We provide more explanation for the solution in Eq. 9. In the first case, $p_g^*(x_i) = 2p_{\mathcal{D}}(x_i)$ if $\sum_{j=0}^{i} p_g^*(x_j) < 1$. That is, if the optimal solution for the primary generator $p_g^*$ *can* assign the probability $2p_{\mathcal{D}}(x_i)$ to the $i^{th}$ in support minima of $f(x)$ without the total sum of probability assigned $\sum_{j=0}^{i} p_g^*(x_j)$ going over 1, then the primary generator $p_g^*$ will assign the probability $2p_{\mathcal{D}}(x_i)$ to $x_i$.

In the second case, $p_g^*(x_i) = 1 - \sum_{j=0}^{i-1} p_g^*(x_j)$ if $\sum_{j=0}^{i} p_g^*(x_j) > 1$. That is, if by assigning the probability $2p_{\mathcal{D}}(x_i)$ to the $i^{th}$ in support minima of $f(x)$, the total sum of probability assigned $\sum_{j=0}^{i} p_g^*(x_j)$ *goes over* 1, then the primary generator $p_g^*$ will assign the remaining probability $1 - \sum_{j=0}^{i-1} p_g^*(x_j)$ to $x_i$. In the third case, the generator assigns probability 0 to $x_i$ because all the probability has already been assigned.

To summarize the benefit of dual generator, we note that by introducing an auxiliary generator and mixing it with the primary generator, not only does the optimal solution for the mixed distribution match the real data distribution, but also the primary generator can better optimize the secondary objective $f$ on the part of the domain of $f$ that is within the support of the data distribution $p_{\mathcal{D}}$. To better illustrate the benefit, we provide a visual explanation of the benefit in Figure 1.

## 4.2 Update rules for offline reinforcement learning

We will now incorporate the dual-generator method to train policies for offline RL, based on optimizing the joint objective from Eq. 5. The updates for the actor and the critic are generally similar to Eq. 3. However, simply combining Eq. 5 and Eq. 3 can still allow the policy to exploit errors in the value function during the policy improvement step. We therefore augment the policy improvement step with an adaptive weight on the Q-value.

More concretely, as the policy improvement step samples actions from the current policy iterate $\pi^k$ to optimize the policy objective, ', as we observe in our experiments. To alleviate this issue, we use the probability assigned to the sampled actions by the discriminator to weight the value function estimates in the policy objective, leading to the following updates:

$$Q^{k+1} \leftarrow \operatorname*{arg\,min}_{Q} \mathbb{E}_{s,a,s' \sim \mathcal{D}} \left[ \left( (R(s,a) + \gamma \mathbb{E}_{a' \sim \pi^k(a'|s')}[Q^k(s',a')]) - Q_{target}(s,a) \right)^2 \right] \tag{10}$$

$$\pi^{k+1} \leftarrow \operatorname*{arg\,max}_{\pi} \mathbb{E}_{s,a_{\mathcal{D}} \sim \mathcal{D}, a \sim \pi^k(a|s)} \left[ \frac{D^k(s,a)}{D^k(s,a_{\mathcal{D}}(s))} Q^{k+1}(s,a) + \log D^k(s,a) \right], \tag{11}$$

where $a_{\mathcal{D}}(s)$ is the action from the offline dataset. The term $D^k(s,a)$ down-weights the contribution of the gradient of the value function to the policy update if the discriminator deems the sampled

action too unlikely. We further calibrate the probability $D^k(s, a)$ by dividing it with the probability $D^k(s, a_{\mathcal{D}}(s))$ that the discriminator assigns to the dataset action $a_{\mathcal{D}}(s)$. It should be noted that during optimization the gradients are not propagated into these weights.

Next, we define the update rules for the auxiliary generator and the discriminator. We mix the samples from the $k^{th}$ iterate of the policy $\pi^k$ and the distribution $p_{aux}$ induced by the $k^{th}$ iterate of the auxiliary generator $G_{aux}^k$, that is, let $p_{mix} = \dfrac{\pi^k + p_{aux}}{2}$. At every iteration $k$, we update the $k^{th}$ iterate of the auxiliary generator $G_{aux}^k$ and discriminator $D^k$ using the objectives:

$$G_{aux}^{k+1} \leftarrow \arg\min_{G_{aux}} \mathbb{E}_{x \sim p_{mix}}[\log(1 - D^k(s, a))] \tag{12}$$

$$D^{k+1} \leftarrow \arg\max_{D} \mathbb{E}_{x \sim p_{\mathcal{D}}}[\log(D^k(s, a))] + \mathbb{E}_{x \sim p_{mix}}[\log(1 - D^k(s, a))] \tag{13}$$

### 4.3 Algorithm summary

Algorithm 1 provides a step-by-step description of our algorithm. At every training step, we sample a batch of transitions from the offline dataset and proceed to update the parameters of the value function, the policy, the auxiliary generator and the discriminator in that order.

---

**Algorithm 1** DASCO algorithm summary

---

1: Initialize Q-function $Q_\theta$, policy $\pi_\phi$, auxiliary generator $G_{aux,\psi}$, discriminator $D_\omega$
2: **for** training step $k$ in $\{1,\dots,N\}$ **do**
3:     $(s, a, r, s') \leftarrow \mathcal{D}$: Sample a batch of transitions from the dataset
4:     $\theta^{k+1} \leftarrow$ Update Q-function $Q_\theta$ using the Bellman update in Eq. 10
5:     $\phi^{k+1} \leftarrow$ Update policy $\pi_\phi$ using the augmented objective in Eq. 11
6:     $\psi^{k+1} \leftarrow$ Update auxiliary generator $G_{aux,\psi}$ using the objective in Eq. 12
7:     $\omega^{k+1} \leftarrow$ Update discriminator $D_\omega$ using mixed samples from $\pi_\phi$ and $G_{aux,\psi}$ as in Eq. 13

---

## 5 Experiments

Our experiments aim at answering the following questions:

1. When learning from offline datasets that require combining actions from sub-optimal trajectories, does DASCO outperform existing methods that are based on distribution constraints?

2. On standard benchmarks such as D4RL [11], how does DASCO compare against recent methods?

3. Are both the dual generator and the probability ratio weight important for the performance of DASCO?

### 5.1 Comparisons on standard benchmarks and new datasets

For our first set of experiments, we introduce four new datasets to simulate the challenges one might encounter when using offline RL algorithms on real world data. These datasets introduce additional learning challenges and require the algorithm to combine actions in different trajectories to obtain good performance. We use the existing AntMaze environments from the D4RL suite [11]: antmaze-medium and antmaze-large. In these two environments, the algorithm controls an 8-DoF "Ant" quadruped robot to navigate a 2D maze to reach desired goal locations. The D4RL benchmark generates the offline datasets for these two environments using two policies: 1. a low-level goal reaching policy that outputs torque commands to move the Ant to a nearby goal location and 2. a high-level waypoint generator to provide sub-goals that guide the low-level goal-reaching policy to the desired location. We use the same two policies to generate two new classes of datasets.

For the `noisy` dataset, we add Gaussian noise to the action computed by the low-level goal-reaching policy. The noise variance depends on the 2D location of the Ant in the maze – larger in some 2D regions than others. We intend this dataset to be representative of situations where the data generation

Table 1: Performance comparison to baselines when learning from the `noisy` and `biased` AntMaze datasets. Our method outperforms the baselines significantly. The value in parenthesis indicates the standard deviation of mean episode return, computed over 3 different runs.

| Dataset | BEAR | EDAC | CQL | IQL | DASCO (Ours) |
|---|---|---|---|---|---|
| antmaze-large-bias | - | - | 61.7 (3.5) | 41.0 (7.9) | 63.9 (6.0) |
| antmaze-large-noisy | - | - | 50.3 (2.3) | 39.0 (6.4) | 54.3 (2.0) |
| antmaze-medium-bias | 0.0 (0.0) | 0.0 (0.0) | 66.7 (2.9) | 48.0 (5.9) | 90.2 (2.4) |
| antmaze-medium-noisy | 0.0 (0.0) | 0.0 (0.0) | 55.7 (4.7) | 44.3 (1.7) | 86.3 (4.5) |
| `noisy` and `biased` antmaze-v2 total | - | - | 234.4 | 172.3 | **294.7** |

Table 2: Performance comparison to distribution-constrained baselines on AntMaze tasks in D4RL. Our algorithm outperforms the baselines when learning from the diverse and play datasets.

| Dataset | CQL | IQL | DASCO (Ours) |
|---|---|---|---|
| antmaze-umaze | 97.0 (0.8) | 90.3 (1.9) | 99.2 (0.0) |
| antmaze-umaze-diverse | 58.7 (12.2) | 70.3 (4.6) | 89.0 (1.7) |
| antmaze-medium-play | 77.0 (1.6) | 82.7 (0.5) | 92.3 (1.5) |
| antmaze-medium-diverse | 80.0 (0.0) | 82.3 (1.9) | 87.1 (0.4) |
| antmaze-large-play | 53.3 (4.6) | 55.7 (3.1) | 64.4 (1.7) |
| antmaze-large-diverse | 48.0 (2.9) | 50.0 (3.6) | 74.1 (2.8) |
| antmaze total | 414.0 | 431.3 | **506.1** |

policies are more deterministic in some states than others [31] – a robot picking up an object has many good options to approach the object, but when the robot grasps the object, its behavior should be more deterministic to ensure successful grasp without damaging or dropping the object [38].

For a `biased` dataset, in addition to adding Gaussian noise to the actions as it is done in the `noisy` dataset, we also add bias to the actions computed by the low-level policy. The values of the bias also depend on the current 2D location of the Ant in the maze. This setting is meant to simulate learning from relabelled data, where the dataset was generated when the data generation policies were performing a different task than the tasks we are using the dataset to learn to perform. Relabelling offline data is a popular method for improving the performance of offline RL algorithms [46, 43], especially when we have much more data for some tasks than others [23]. In the AntMaze environment, offline RL algorithms must combine data from sub-optimal trajectories to learn behaviors with high returns. In addition, `noisy` and `biased` datasets present a more challenging learning scenarios due to the added noise and systematic bias which vary non-uniformly based on the 2D location of the Ant.

Table 1 illustrates the performance comparison of our method and representative methods that enforce distribution constraints, either through optimizing a conservative lower bound of the value estimates (CQL) or only optimizing the policy on actions in the dataset using Advantage Weighted Regression [41] (IQL). Our method outperforms both CQL and IQL. In these tasks, to ensure a fair comparison between different methods, we perform oracle offline policy selection to obtain the performance estimates for CQL, IQL, and our method. We describe how we tune hyper-parameters for the baselines in Appendix E. We also include results of recent methods, such as EDAC [2] and BEAR [28]. We also compare the performance on standard AntMaze tasks when learning from the datasets in the D4RL benchmark without modifications in Table 2. In these tasks, our method outperforms IQL by a large margin on two diverse datasets.

By comparing the results in Table 1 (learning from `noisy` and `biased` datasets) and Table 2 (learning from existing offline datasets in D4RL), we also note that our proposed algorithm outperforms distribution-constraint offline RL algorithms (CQL, IQL) more consistently when tested on the `noisy` and `biased` datasets. For the results in these two tables, the definition of the antmaze-medium and antmaze-large environments are the same. The only axis of variation in the learning setup is the noise and systematic bias added to the actions of the dataset generation policies. We therefore conclude that our algorithm is more robust to the noise and systematic bias added to the actions than distribution-constrained offline RL algorithms.

Next, we evaluate our approach on Gym locomotion tasks from the standard D4RL suite. The performance results on these tasks are illustrated in Table 3. Our method is competitive with BC,

Table 3: Performance comparison with recent offline RL algorithms on the Gym locomotion tasks

| Dataset | BC | 10%BC | DT [7] | AWAC [39] | Onestep RL [6] | TD3+BC [13] | COMBO [50] | CQL | IQL | DASCO (Ours) |
|---|---|---|---|---|---|---|---|---|---|---|
| halfcheetah-medium-replay | 36.6 | 40.6 | 36.6 | 40.5 | 38.1 | 44.6 | 55.1 | 45.5 | 44.2 | 44.7 |
| hopper-medium-replay | 18.1 | 75.9 | 82.7 | 37.2 | 97.5 | 60.9 | 89.5 | 95.0 | 94.7 | 101.7 |
| walker2d-medium-replay | 26.0 | 62.5 | 66.6 | 27.0 | 49.5 | 81.8 | 56.0 | 77.2 | 73.9 | 74.5 |
| halfcheetah-medium-expert | 55.2 | 92.9 | 86.8 | 42.8 | 93.4 | 90.7 | 90.0 | 91.6 | 86.7 | 93.8 |
| hopper-medium-expert | 52.5 | 110.9 | 107.6 | 55.8 | 103.3 | 98.0 | 111.1 | 105.4 | 91.5 | 110.9 |
| walker2d-medium-expert | 107.5 | 109.0 | 108.1 | 74.5 | 113.0 | 110.1 | 103.3 | 108.8 | 109.6 | 109.3 |
| locomotion total | 295.9 | 491.8 | 488.4 | 277.8 | 494.8 | 486.1 | 505 | 523.5 | 500.6 | 534.9 |

Table 4: Ablation for training without and with auxiliary generator. The dual generator technique, which trains the auxiliary generator in addition to the policy, is crucial to obtain good performance.

| Dataset | Without | With |
|---|---|---|
| halfcheetah-medium-expert | 79.8 (3.4) | 93.8 (0.1) |
| hopper-medium-expert | 95.1 (1.6) | 110.9 (0.8) |
| antmaze-large-bias | 55.0 (2.3) | 63.9 (6.0) |
| antmaze-large-noisy | 45.1 (1.8) | 54.3 (2.0) |

one-step offline RL methods [5], and multi-step distribution-constraint RL methods [26, 30]. This is not surprising because in these tasks, the offline dataset contains a large number of trajectories with high returns. In Table 3, 10%BC refers to performing BC using the top 10% trajectories with the highest episode return.

## 5.2 Ablations

We conduct three different sets of experiments to gain more insights into our algorithm. The first experiment measures the importance of having an auxiliary generator. We recall that there are two benefits to having the auxiliary generator. Firstly, without the auxiliary generator, the generator does not in general match the data distribution (Theorem 4.1). As such, the discriminator has an unfair advantage in learning how to distinguish between real and generated examples. Secondly, the auxiliary generator plays the role of a support player and learns to output actions that are assigned non-zero probability by the data distribution, but have low Q values. The support player allows the policy to concentrate on in-support maximization of the Q-function (Theorem 4.2). Table 4 demonstrates that having an auxiliary generator clearly leads to a performance improvement across different task families, from Gym locomotion tasks to AntMaze navigation tasks.

The second experiment compares the performance of the policy and the auxiliary generator on a subset of the Gym locomotion and AntMaze tasks (Table 5). The difference in the performance of the policy and auxiliary generator illustrates their specialization of responsibility: the policy learns to output actions that lead to good performance, while the auxiliary generator learns to model the "remainder" of the data distribution. If this "remainder" also contains good action, then the auxiliary generator will have non-trivial performance. Otherwise, the auxiliary generator will have poor performance.

In the Gym locomotion tasks, the auxiliary generator has non-trivial performance, but it is still worse than the policy. This demonstrates that: 1. By optimizing the policy to maximize the long-term return and the discriminator function, the policy can outperform the auxiliary generator, which only maximizes the discriminator function, 2. The dataset contains a large fraction of medium performance level actions contained in continuous trajectories, which the auxiliary generator has learnt to output. In contrast, in the `bias` and `noisy` AntMaze tasks, the auxiliary generator fails to obtain non-zero performance while the policy has strong performance. This reflects the necessity of carefully picking a subset of the in-support actions to obtain good performance.

The third set of experiments illustrates the importance of weighing the value function in the policy objective by the probability computed by the discriminator, as described in Eq. 11. Doing so provides a second layer of protection against exploitation of errors in the value function by the policy. Table 6 illustrates that this is very important for the AntMaze tasks, which require combining optimal and sub-optimal trajectories to obtain good performance. Perhaps this is because learning from such trajectories necessitates many rounds of offline policy evaluation and improvement steps, with each round creating an opportunity for the policy to exploit the errors in the value estimates. On the other

Table 5: Policy vs Auxiliary Generator. The auxiliary generator has reasonable performance on the easier locomotion tasks and is significantly worse than the policy on the harder AntMaze tasks.

| Dataset | Auxiliary Generator | Policy |
|---|---|---|
| halfcheetah-medium-expert | 48.5 (2.1) | 93.8 (0.1) |
| hopper-medium-expert | 70.4 (0.9) | 110.9 (0.8) |
| antmaze-large-bias | 0.0 (0.0) | 63.9 (6.0) |
| antmaze-large-noisy | 0.0 (0.0) | 54.3 (2.0) |

Table 6: Ablation for dynamic weighting of value function estimates in the policy objective. When learning from datasets that require combining actions across trajectories, such as the AntMaze tasks, using the dynamic weighting is vital to obtaining good performance.

| Dataset | Without | With |
|---|---|---|
| halfcheetah-medium-expert | 91.1 (1.1) | 93.8 (0.1) |
| hopper-medium-expert | 106.7 (2.9) | 110.9 (0.8) |
| antmaze-large-play | 0.0 (0.0) | 64.4 (1.7) |
| antmaze-large-diverse | 0.0 (0.0) | 74.1 (2.8) |

hand, the dynamic weight is less important in the Gym locomotion tasks, presumably because a significant fraction of the corresponding offline datasets has high returns and therefore incorporating sub-optimal data is less criticial to obtain high performance.

## 6   Conclusions

In this paper, we introduced DASCO, a GAN-based offline RL method that addresses the challenges of training policies as generators with a discriminator to minimize deviation from the behavior policy by means of two modifications: an auxiliary generator to turn the GAN loss into a support constraint, and a value function weight in the policy objective. The auxiliary generator makes it possible for the policy to focus on maximizing the value function without needing to match the *entirety* of the data distribution, only that part of it that has high value, effectively turning the standard distributional constraint that would be enforced by a conventional GAN into a kind of support constraint. This technique may in fact be of interest in other settings where there is a need to maximize some objective in addition to fooling a discriminator, and applications of this approach outside of reinforcement learning are an exciting direction for future work. Further, since our method enables GAN-based strategies to attain good results on a range of offline RL benchmark tasks, it would also be interesting in future work to consider other types of GAN losses that induce different divergence measures. We also plan to explore robust methods for offline policy and hyper-parameter selection in the future.

**Potential negative societal impact** Because our focus is on developing a generic offline RL algorithm and not its application in any particular domain, our method inherits the potential negative societal impacts that any offline RL algorithm might have, such as data bias and lack of robustness to distributional shift. We also note that how to perform effective offline policy selection and evaluation in the general case remain challenging. Therefore, the offline learned RL policies might not be easily deployed to solve real world problems where online policy evaluation is expensive.

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
