# Appendices

## A Proofs for theorems in Section 4.1

### A.1 Proof for Theorem 4.1

In the following proof, we use $p_{\text{data}}$ to refer to the real data distribution, instead of $p_{\mathcal{D}}$ as in Section 4.1, to avoid confusion with the discriminator distribution.

We recall Theorem 4.1:

**Theorem 4.1** *The optimal generator of Eq. 4 induces a distribution* $p_g^*(x) = p_{\mathcal{D}}(x)\dfrac{e^{-f(x)-\nu}}{2 - e^{-f(x)-\nu}}$, *where* $\nu > 0$ *is the Lagrange multiplier that ensures that* $p_g^*(x)$ *is normalized to 1.*

The optimization problem in Eq. 4 is:

$$\min_G \max_D V(G, D) = \mathbb{E}_{x \sim p_{\text{data}}}[\log(D(x))] + \mathbb{E}_{z \sim p(z)}[\log(1 - D(G(z)))] + \mathbb{E}_{z \sim p(z)}[f(G(z))]$$

The proof proceeds as follows: We first simplify the objective function into two terms. The first term is the Jensen–Shannon divergence between the data distribution and the distribution induced by the generator [16]. The second term is the expected value of the secondary objective function $f$. We then show that the problem is convex, where strong duality holds. We then use the KKT conditions to find the functional form of the optimal solution, which gives us Theorem 4.1.

We only prove the statement for discrete sample space, and we let $n$ be the size of the sample space – the random variable $x$ can take on $n$ different values.

*Proof.* Since the third term in the objective function is not a function of the discriminator $D$, for $G$ fixed, the optimal discriminator of Eq. 4 is $D_G^*(x) = \dfrac{p_{\text{data}}(x)}{p_{\text{data}}(x) + p_g(x)}$ where $p_g$ is the distribution induced by the generator $G$. (similar to Prop 1 in [16] ).

Similarly to how [16] shows that the GAN objective in Eq. 1 minimizes the JS divergence between the data distribution and the distribution induced by the generator, we can now rewrite the objective in Eq. 4 as:

$$V(G, D_G^*) \tag{14}$$
$$= \mathbb{E}_{x \sim p_{\text{data}}}[\log(D_G^*(x))] + \mathbb{E}_{z \sim p(z)}[\log(1 - D_G^*(G(z)))] + \mathbb{E}_{z \sim p(z)}[f(G(z))] \tag{15}$$
$$= 2JSD(p_{\text{data}}||p_g) + \mathbb{E}_{x \sim p_g}[f(x)] - \log 4 \tag{16}$$

For conciseness, let $g^{(i)} = p_g(x_i)$ be the probability that $p_g$ assigns to $x_i$ and $g = [g^{(1)}, \ldots, g^{(n)}]^T$ be a column vector containing the probabilities that $p_g$ assigns to each possible values of $x$, from $x_1$ to $x_n$.

Similarly, let $f^{(i)} = f(x_i)$ be the value that the secondary objective $f$ assigns to $x_i$. We also overload the notation to let $f = [f^{(1)}, \ldots, f^{(n)}]^T$ be a column vector containing the values that the secondary objective $f$ assigns to each possible value of the random variable $x$, from $x_1$ to $x_n$.

Also let $p_{\text{data}}^{(i)} = p_{\text{data}}(x_i)$ be the probability that the data distribution assigns to $x_i$.

We can then rewrite the problem in Eq. 4 in a standard form [4] as:

$$\min_g \quad 2JSD(p_{\text{data}}||p_g) + g^T f \tag{17}$$

$$\text{s.t.} \quad -g^{(i)} \leq 0 \tag{18}$$

$$\mathbf{1}^T g - 1 = 0 \tag{19}$$

where $\mathbf{1}$ is a column vector of 1, which has the same number of entries as the vector $g$. The constraint 18 ensures that the probability that $p_g$ assigns to any $x$ is non-negative and the constraint 19 ensures the probabilities sum up to 1.

The problem is convex because the objective function is a nonnegative weighted sum of two convex functions (JSD is convex because JSD is itself a nonnegative weighted sum of KL, which is a convex function).

Strong duality also holds because Slater's condition holds. A strictly feasible point for Slater's condition to hold is the uniform distribution, i.e. $g^{(i)} = \dfrac{1}{n}, \forall i$.

The Lagrangian is:

$$L = 2JSD(p_{\text{data}}||p_g) + g^T f - \sum_i \lambda^{(i)} g^{(i)} + \nu(\mathbf{1}^T g - 1) \tag{20}$$

where $\lambda^{(i)}$ and $\nu$ are the Lagrangian multipliers.

For any $i \in [1, n]$, the partial derivative of the Lagrangian with respect to $g^{(i)}$ is:

$$\frac{\partial L}{\partial g^{(i)}} = log\left(\frac{2g^{(i)}}{p_{\text{data}}^{(i)} + g^{(i)}}\right) + f^{(i)} - \lambda^{(i)} + \nu \tag{21}$$

Let $g_*$ and $(\lambda_*, \nu_*)$ be the primal and dual optimal solutions of the optimization problem. As the strong duality holds, the variables $g_*$ and $(\lambda_*, \nu_*)$ must satisfy the KKT conditions. For any $i \in [1, n]$, the following holds:

$$-g_*^{(i)} \leq 0 \tag{22}$$

$$\mathbf{1}^T g_* - 1 = 0 \tag{23}$$

$$\lambda_*^{(i)} \geq 0 \tag{24}$$

$$\lambda_*^{(i)} g_*^{(i)} = 0 \tag{25}$$

$$\frac{\partial L}{\partial g^{(i)}} = log\left(\frac{2g_*^{(i)}}{p_{\text{data}}^{(i)} + g_*^{(i)}}\right) + f^{(i)} - \lambda_*^{(i)} + \nu_* = 0 \tag{26}$$

From Equation 26, we have $\lambda_*^{(i)} = log\left(\dfrac{2g_*^{(i)}}{p_{\text{data}}^{(i)} + g_*^{(i)}}\right) + f^{(i)} + \nu_*$, and substitute into Equation 25:

$$\left[log\left(\frac{2g_*^{(i)}}{p_{\text{data}}^{(i)} + g_*^{(i)}}\right) + f^{(i)} + \nu_*\right] g_i^* = 0 \tag{27}$$

We consider what happens when $g_i^* > 0$, due to complementary slackness, we have:

$$log\left(\frac{2g_*^{(i)}}{p_{\text{data}}^{(i)} + g_*^{(i)}}\right) + f^{(i)} + \nu_* = 0 \tag{28}$$

$$\implies g_*^{(i)} = \frac{p_{\text{data}}^{(i)} e^{-f^{(i)} - \nu_*}}{(2 - e^{-f^{(i)} - \nu_*})} \tag{29}$$

$$p_g^*(x_i) = p_{\text{data}}(x_i)\frac{e^{-f(x_i) - \nu_*}}{2 - e^{-f(x_i) - \nu_*}} \tag{30}$$

We can then pick an appropriate value for the Lagrange multiplier $\nu$ such that the probabilities $p_g^*(x_i)$ normalize to 1. QED.

## A.2    Proof for Theorem 4.2

In the following proof, we use $p_{\text{data}}$ to refer to the real data distribution, instead of $p_{\mathcal{D}}$ as in Section 4.1, to avoid confusion with the discriminator distribution.

Recall that we define $p_{mix}$ as $p_{mix} = \dfrac{p_g + p_{aux}}{2}$. Theorem 4.2 is stated in reference to the optimization problem in Eq. 5, which we restate here:

$$\min_{G,G_{aux}} \max_D \quad V(G, G_{aux}, D) = \mathbb{E}_{x \sim p_{\text{data}}}[\log(D(x))] + \mathbb{E}_{x \sim p_{mix}}[\log(1 - D(x))] + \mathbb{E}_{x \sim p_g}[f(x)] \tag{31}$$

where the first two terms in the objective function are the GAN objective and the last term is the secondary objective function.

Similar to the proof for Theorem 4.1, we can rewrite the objective function in Eq. 31 as [16]:

$$V(G, G_{aux}, D^*) \tag{32}$$

$$= 2JSD(p_{\text{data}} || \frac{p_g + p_{aux}}{2}) + \mathbb{E}_{x \sim p_g}[f(x)] - \log 4 \tag{33}$$

We are only interested in optimizing for the secondary objective function $f$ in the space of optimal GAN solutions. We therefore enforce that $p_{mix} = \dfrac{p_g + p_{aux}}{2} = p_{\text{data}}$, which makes the JSD term vanish in Eq. 33 and allows us to solve the following optimization problem.

$$\min_G \quad \mathbb{E}_{x \sim p_g}[f(x)] \tag{34}$$

$$\text{s.t.} \quad p_g \le 2p_{\text{data}} \tag{35}$$

$$p_{aux} = 2p_{\text{data}} - p_g \tag{36}$$

We claim that the solution to the optimization problem above is as follows. We define $x_0$ to be the element inside the support of the data distribution $p_{\text{data}}$ that minimizes $f$, i.e. $x_0 = \underset{x \in \text{Supp}(p_{\text{data}})}{\arg\min} f(x)$.

The optimal primary generator $p_g^*$ assigns the following probability to $x_0$:

$$p_g^*(x_0) = \begin{cases} 2p_{\text{data}}(x_0) & \text{if} \quad 2p_{\text{data}}(x_0) < 1 \\ 1 & \text{otherwise} \end{cases} \tag{37}$$

If the global maximum $x_0$ is not taking the full probability mass, the rest of the probability mass is redistributed to the next best in-support maxima, which we can define recursively:

$$\text{For } x_i \in \underset{x \in \text{Supp}(p_{\text{data}}) \setminus \{x_j\}_{j=0}^{i-1}}{\arg\min} f(x), \ p_g^*(x_i) = \begin{cases} 2p_{\text{data}}(x_i) & \text{if} \quad \sum_{j=0}^{i} p_g^*(x_j) < 1 \\ 1 - \sum_{j=0}^{i-1} p_g^*(x_j) & \text{if} \quad \sum_{j=0}^{i} p_g^*(x_j) > 1 \\ 0 & \text{if} \quad \sum_{j=0}^{i-1} p_g^*(x_j) = 1 \end{cases} \tag{38}$$

*Proof.*

We show the proof by contradiction. That is, assume that there exists another distribution $p_g^a$ with the following properties:

- There exists $x$ where $p_g^a(x) \ne p_g^*(x)$
- $p_g^a$ satisfies the constraint (35)-(36)
- The value of the objective function achieved by $p_g^a$ is better than the value achieved by $p_g^*$. That is, $\mathbb{E}_{x \sim p_g^a}[f(x)] < \mathbb{E}_{x \sim p_g^*}[f(x)]$.

We will show that the existence of such a distribution $p_g^a$ will lead to contradiction,

We separate the analyses into three different cases, depending on the property of $p_g^*$:

- Case 1: $p_g^*$ assigns all probability mass to $x_0$
- Case 2: If $p_g^*$ assigns non-zero probability to $x$, then $p_g^* = 2p_{\text{data}}(x)$

- Case 3: There exists an $x$ where $2p_{\text{data}}(x) > p_g^*(x) > 0$

We will walk through the three cases independently and show the contradiction in each case.

**Case 1:** $p_g^*$ assigns the full probability mass to $x_0$, that is $p_g^*(x_0) = 1$, and assigns zero probability to every $x$ not equal to $x_0$. Without loss of generality, we consider $p_g$ that assigns non-zero probability to a $x_k \neq x_0$, assigns the remaining probability mass to $x_0$, and assigns zero probability to all $x$ that is not equal to either $x_0$ or $x_k$. That is, assume there exists $p_g^a$ such that:

$$0 > p_g^a(x_0) > 1 \tag{39}$$

$$p_g^a(x_k) = 1 - p_g^a(x_0) > 0 \text{ for some } x_k \in \text{Supp}(p_{\text{data}}) \tag{40}$$

$$\mathbb{E}_{x \sim p_g^*}[f(x)] - \mathbb{E}_{x \sim p_g^a}[f(x)] > 0 \tag{41}$$

where $x_k \in \text{Supp}(p_{\text{data}})$ follows from constraint 35 ($p_g \leq 2p_{\text{data}}$, and thus $p_g^a$ can only assign non-zero probability to $x$ within the support of $p_{\text{data}}$). We can then show that:

$$\mathbb{E}_{x \sim p_g^*}[f(x)] - \mathbb{E}_{x \sim p_g^a}[f(x)] \tag{42}$$

$$= f(x_0) - p_g^a(x_0)f(x_0) - p_g^a(x_k)f(x_k) \tag{43}$$

$$= (1 - p_g^a(x_0))f(x_0) - p_g^a(x_k)f(x_k) \tag{44}$$

$$= p_g^a(x_k)f(x_0) - p_g^a(x_k)f(x_k) \tag{45}$$

$$= p_g^a(x_k)[f(x_0) - f(x_k)] \leq 0 \text{ (contradiction with Eq.41)} \tag{46}$$

where the last inequity follows from these two facts:

$$x_0 = \underset{x \in \text{Supp}(p_{\text{data}})}{\arg \min} f(x) \tag{47}$$

$$x_k \in \text{Supp}(p_{\text{data}}) \tag{48}$$

**Case 2:**

$$p_g^*(x) = \begin{cases} 2p_{\text{data}}(x) & \text{if } p_g^*(x) > 0 \\ 0 & \text{otherwise} \end{cases} \tag{49}$$

Let $\{x_0, \ldots, x_i\}$ be the set of x where $p_g^*(x) > 0$, then we also require that $\sum_{j=0}^{i} p_g^*(x) = 1$.

Without loss of generality, we assume a distribution $p_g^a$ exists with the following properties. There exists $x_m, x_n$ such that:

$$p_g^*(x_m) = 2p_{\text{data}}(x_m) > 0 \quad \text{and} \quad p_g^a(x_m) < 2p_{\text{data}}(x_m) \tag{50}$$

$$p_g^*(x_n) = 0 \quad \text{and} \quad p_g^a(x_n) = 2p_{\text{data}}(x_m) - p_g^a(x_m) > 0 \tag{51}$$

$$p_g^*(x) = p_g^a(x) \text{ otherwise (that is, for all } x \notin \{x_m, x_n\}) \tag{52}$$

$$\mathbb{E}_{x \sim p_g^*}[f(x)] - \mathbb{E}_{x \sim p_g^a}[f(x)] > 0 \tag{53}$$

We note that $f(x_m) \leq f(x_n)$ since $p_g^*$ assigns non-zero probability to $x_m$ and assigns zero probability to $x_n$.

We can show that:

$$\mathbb{E}_{x \sim p_g^*}[f(x)] - \mathbb{E}_{x \sim p_g^a}[f(x)] \tag{54}$$

$$= p_g^*(x_m)f(x_m) - p_g^a(x_m)f(x_m) - p_g^a(x_n)f(x_n) \tag{55}$$

$$= p_g^*(x_m)f(x_m) - p_g^a(x_m)f(x_m) - p_g^a(x_n)f(x_n) \tag{56}$$

$$= p_g^*(x_m)f(x_m) - p_g^a(x_m)f(x_m) - (2p_{\text{data}}(x_m) - p_g^a(x_m))f(x_n) \tag{57}$$

$$= p_g^*(x_m)f(x_m) - p_g^a(x_m)f(x_m) - 2p_{\text{data}}(x_m)f(x_n) + p_g^a(x_m)f(x_n) \tag{58}$$

$$= p_g^*(x_m)f(x_m) - p_g^a(x_m)f(x_m) - p_g^*(x_m)f(x_n) + p_g^a(x_m)f(x_n) \tag{59}$$

$$= p_g^*(x_m)[f(x_m) - f(x_n)] - p_g^a(x_m)[f(x_m) - f(x_n)] \tag{60}$$

$$= [f(x_m) - f(x_n)][p_g^*(x_m) - p_g^a(x_m)] \leq 0 \text{ (contradiction with Eq.53)} \tag{61}$$

where the last inequality is true because $f(x_m) \leq f(x_n)$ as we noted above, and $p_g^*(x_m) = 2p_{\text{data}}(x_m) > p_g^a(x_m)$.

**Case 3:**

There exists $x_i$ such that $2p_{\text{data}}(x_i) > p_g^*(x_i) > 0$. For all $x \neq x_i$:

$$p_g^*(x) = \begin{cases} 2p_{\text{data}}(x) & \text{if} \quad p_g^*(x) > 0 \\ 0 & \text{otherwise} \end{cases} \tag{62}$$

Let $\{x_0, \ldots, x_i\}$ be the set of x where $p_g^*(x) > 0$, we also require $\sum_{j=0}^{i} p_g^*(x) = 1$.

Without loss of generality, there are three cases we need to consider for the distribution $p_g^a$, each yielding a contradiction:

- $p_g^a(x_i) = p_g^*(x_i)$, but there exists $x$ such that $p_g^a(x) \neq p_g^*(x)$.
- $p_g^a(x_i) > p_g^*(x_i)$.
- $p_g^a(x_i) < p_g^*(x_i)$.

In each case, the proof by contradiction is similar to the proof in Case 2 above, where we pick a pair of $x_m, x_n$ and shows that $p_g^a$ can not achieve a lower value of the objective function than $p_g^*$. We thus do not repeat the argument here. QED

# B    Benefits of dual generator technique on 1D discrete example

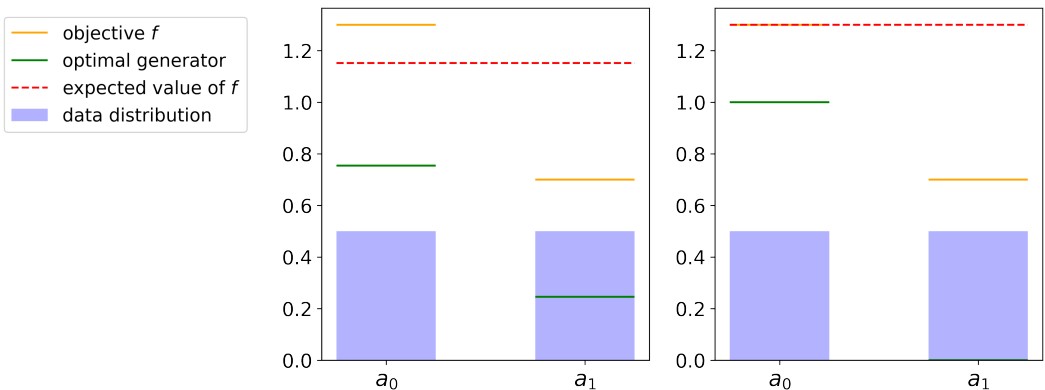

Figure 2: **Left:** We use a single generator. **Right:** We use dual generator technique. The optimal generator (green bar) refers to the optimal primary generator, and not the auxiliary generator. Thanks to the dual generator technique, the optimal primary generator in the right figure assigns probability 1.0 to the global maxima $a_0$ of the secondary objective function $f$. The optimal primary generator in the right figure is therefore better at maximizing the function $f$ than the optimal generator in the left figure. Note that in this example, the generator aims to maximize the function $f$ (instead of minimize) for more intuitive interpretation.

In this section, we provide a simple one-dimensional numerical example with discrete action space to illustrate the benefit of the dual generator technique. In this example, the action space only consists two actions $a_0, a_1$. The probability of action $a_0$ under the data distribution is $0.5$. The probability of action $a_1$ under the data distribution is $0.5$. We would like to maximize a secondary objective function $f$. The function $f$ assigns value $1.3$ to $a_0$ and value $0.7$ to $a_1$.

We will next show in a self-contained Jupyter notebook that when using only a single generator, the expected value of the secondary objective function $f$ under the optimal generator is $1.15$. We also show that when using the dual generator technique, the expected value of the secondary objective function $f$ under the optimal primary generator is $1.3$. Since $1.3$ is higher than $1.15$, we can see that using the dual generator technique allows us to better maximize the objective function $f$.

Fig 2 visually illustrates the benefits of the dual generator technique over using only a single generator in this example. We can also observe from Fig 2 (Left) that the optimal generator is clearly different from the data distribution when only using a single generator. As such, in the GAN framework, the discriminator has an advantage in learning how to distinguish between samples from the real data distribution and samples from the generator. In contrast, when using the dual generator technique, the optimal primary generator assigns probability 1 to action $a_0$ and the optimal auxiliary generator assigns probability 1 to action $a_1$ (not shown in Fig 2). Their mixed distribution therefore assigns probability $0.5$ to either actions, matching the data distribution.

The Jupyter notebook in the next page illustrates the computations necessary to obtain the optimal solution for the generator.

```
In [1]:  import numpy as np
         from scipy.optimize import fsolve
```

In this toy example, the action space consists of two actions, denoted by $a_0$ and $a_1$. The probability of action $a_0$ under the data distribution is $0.5$. The probability of action $a_1$ under the data distribution is $0.5$. The data distribution is represented by the variable $p_{data}$ in the code cell below.

```
In [2]:  p_data = np.array([
             0.5, 0.5
         ])
```

The secondary objective function that we would like to **maximize** is represented by $f$, where $f(a_0) = 1.3$ and $f(a_1) = 0.7$.

```
In [3]:  f = np.array([
             1.3, 0.7
         ])
```

When using only a single generator and not employing the dual generator technique, we can write the closed form solution of the optimal generator as a function of the data distribution $p_{data}$, the secondary objective $f$, and a Lagrange variable $nu$ as follows. The optimal solution of the generator is represented by the variable $p\_g\_star$ in the code cell below:

```
In [4]:  def get_p_star_single_generator(p_data, f, nu):

             # We use f instead of -f because
             # we are maximizing f in this example.
             exp = np.exp( f - nu )

             p_g_star = p_data * ( exp / ( 2 - exp ) )

             return p_g_star
```

Since the distribution induced by the optimal generator must be a valid distribution, we solve for the Lagrange variable $nu$ such that the probabilities that the optimal generator assigns to the two actions $a_0, a_1$ sum to $1$.

```
In [5]:  def func(nu):

             # p_g_star is a array consisting of two elements.
             # The first element is the density that the optimal
             # generator assigns to the action a_0.
             # The second element is the density that the optimal
             # generator assigns to the action a_1.
             p_g_star = get_p_star_single_generator(p_data, f, nu)

             sum_p_g_star = p_g_star.sum()

             return 1.0 - sum_p_g_star

         # nu_star is the value of nu that makes the distribution
         # induced by the optimal generator a valid distribution.
         nu_star = fsolve(
             func, 1.
         )
         print('nu_star is', nu_star)

         nu_star is [1.11566684]
```

```
In [6]:   # Confirm that the probabilities sum to 1
          p_g_star = get_p_star_single_generator(p_data, f, nu_star)

          print('The probabilities that the optimal generator assigns to the two actions are:')
          print(p_g_star)

          print('Sum of probabilities', p_g_star.sum().round(2))
```

```
The probabilities that the optimal generator assigns to the two actions are:
[0.75378698 0.24621302]
Sum of probabilities 1.0
```

```
In [7]:   E_f = (p_g_star * f).sum()
          print(
              'Expected value of f under the optimal generator when using a single generator is:'
          )
          print(E_f.round(2))
```

```
Expected value of f under the optimal generator when using a single generator is:
1.15
```

We next derive the optimal solution of the generator when using the dual generator technique and show that using the dual generator technique allows us to better maximize the secondary objective function $f$. When using the dual generator technique, we maintain two generators, a primary generator denoted by $p_g$ and an auxiliary generator denoted by $p_{aux}$.

The in-support global maxima of $f$ is $a_0$, since $f(a_0) = 1.3 > f(a_1) = 0.7$. As such, the optimal solution of the primary generator, denoted by $p_g^*$, would like to assign as much probability to $a_0$ as possible. Since we have the constraint $p_g^*(a_0) \leq 2p_{data}(a_0) = 1.0$, we have that $p_g^*(a_0) = 1.0$. That is, the optimal primary generator assigns probability $1.0$ to action $a_0$. Since there are only two actions, the optimal primary generator assigns probability $0.0$ to the remaining action $a_1$.

The optimal primary generator is represented by the variable $dual\_p\_g\_star$ in the code cell below:

```
In [8]:   dual_p_g_star = np.array([
              1.0, 0.0
          ])

          dual_E_f = (dual_p_g_star * f).sum()
          print(
              'Expected value of f under the optimal generator when using dual generator is:')
          print(dual_E_f.round(2))
```

```
Expected value of f under the optimal generator when using dual generator is:
1.3
```

Since we want to **maximize** the secondary objective function $f$, and $1.3$ is higher than $1.15$, we can see that using the dual generator technique allows us to better maximize the secondary objective function $f$.

## C   Description of the offline dataset generation procedure for the `noisy` and `biased` AntMaze datasets

In the experiments section, we introduce the `bias` and `noisy` datasets for the AntMaze tasks. In this section, we provide more details on how the datasets were generated in the form of Python syntax in Code Listing 1. We plan to open-source both the datasets and the code to generate the datasets upon acceptance.

Code Listing 1: Illustration of the dataset generation procedure for the `bias` and `noisy` datasets. Given an `action` computed by the `behavior_policy`, we add noise and bias to the action. The magnitudes of the noise and bias depend on the x-values of the position of the Ant in the 2D maze.

```
NOISES = [0.1, 0.0, 0.2, 0.05, 0.3, 0.1, 0.4, 0.2]
BIASES = [0.1, -0.1, 0.2, 0.0, 0.2, -0.3, 0.2, 0.0]
POSITION = [-20.0, 0.0, 4.0, 8.0, 12.0, 16.0, 20.0, 24.0]

action = behavior_policy.get_action(obs)

x_position = get_x_position(obs)

pos = [idx for idx in range(len(POSITION)) if POSITION[idx] <=
                                    x_position]
pos = max(pos)

noise = NOISES[pos]
bias = BIASES[pos]

action = action + np.random.normal(size=action.shape) * noise - bias *
                                    np.ones_like(action)
action = np.clip(action, -1.0, 1.0)
```

## D   Additional experimental details

For all tasks, we average mean returns over 20 evaluation trajectories. Similar to the pre-processing steps in previous works [25], we standardize MuJoCo locomotion task rewards by dividing by the difference of returns of the best and worst trajectories in each dataset. For the AntMaze datasets, we subtract 1 from rewards for all transitions. We use Adam optimizer [24] with a learning rate of 0.0003. For the value functions, we use an MLP with 3 hidden layers of size 256. For both the GAN discriminator and auxiliary generator, we use an MLP with 1 hidden layer of size 750. The auxiliary generator takes a state as an input, and a noise vector and output actions deterministically as a function of the input state and noise vector. For the policy, which is also the primary generator, we use an MLP with 4 hidden layers of size 256. The policy takes a state as an input and outputs the parameters of a diagonal Gaussian, from which we sample an action. We update the target network with soft updates with parameter 0.005.

For the discriminator loss function, we use the mean-squared error loss, inspired by LSGAN [36]. For the auxiliary generator, we use the standard vanilla GAN loss. The loss functions and how they are used are further illustrated in Section F. We also use instance noise [45] where we sample the instance noise from a Gaussian distribution for each action dimension independently. The Gaussian is zero-center and has an initial standard deviation of 0.3 at the beginning of training. We anneal the magnitude of the noise over time and also clamp the instance noise to have a maximum magnitude of 0.3. We also train the discriminator for more steps than the generators in each algorithm step (5 training steps for the discriminator for every step of the generator).

In the policy objective (Eq. 11), we also use a hyper-parameter $w$ to weight the contribution of the value function and the discriminator probability to the policy update. That is, we use Eq. 63 to update the policy. We fix the value of $w$ throughout training. For the AntMaze tasks, we set $w = 0.025$. For the Mujoco locomotation task, we set $w = 1.0$.

$$\pi^{k+1} \leftarrow \arg\max_{\pi} \mathbb{E}_{s, a_{\mathcal{D}} \sim \mathcal{D}, a \sim \pi^k(a|s)} \left[ \frac{1}{w} \frac{D^k(s, a)}{D^k(s, a_{\mathcal{D}}(s))} Q^{k+1}(s, a) + \log D^k(s, a) \right], \quad (63)$$

In terms of total amount of compute and type of resources used, we use an internal cluster that allows for access up to 64 preemptive Nvidia RTX 2080 Ti GPUs. For each experiment of learning from an offline dataset, we use half a GPU and 3 CPU cores. Each experiment generally takes half a day to finish. We implemented our algorithms in Pytorch [40].

# E    Additional details on baselines

**IQL** To obtain the result of IQL when learning from the `noisy` and `biased` AntMaze datasets presented in Table 1, we tune the expectile hyper-parameter $\tau$ in IQL. Table 7 illustrates the different values of $\tau$ we ran IQL for and the performance of each value.

Table 7: Performance on IQL when learning from the `noisy` and `biased` AntMaze datasets for different expectile value tau. tau=0.9 performs the best in the large maze, whereas tau=0.95 perform the best in the medium size maze. We therefore present the result using tau=0.9 in Table 1. For lower tau values such as $0.8, 0.7$, the performance is worse. This is expected and also mentioned in Section 5.2 of the IQL paper.

| Dataset | tau=1.0 | tau=0.95 | tau=0.9 | tau=0.8 | tau=0.7 |
|---|---|---|---|---|---|
| antmaze-large-bias | 0.0 (0.0) | 22.0 (0.8) | **41.0** (7.9) | 20.0 (3.6) | 4.0 (1.4) |
| antmaze-large-noisy | 0.0 (0.0) | 37.7 (5.2) | **39.0** (6.4) | 14.0 (5.9) | 3.3 (2.1) |
| antmaze-medium-bias | 0.0 (0.0) | **62.3** (9.8) | 48.0 (5.9) | 43.0 (7.1) | 21.0 (2.9) |
| antmaze-medium-noisy | 0.0 (0.0) | **53.7** (9.0) | 44.3 (1.7) | 41.7 (6.1) | 19.3 (2.5) |

**CQL** We also tune the Lagrange threshold in CQL when learning from the `noisy` and `biased` AntMaze datasets. We refer to this hyper-parameter also as tau (but this is different from the hyper-parameter tau in IQL). The results for different values of the Lagrange threshold tau for CQL are illustrated in Table 8.

Table 8: Performance of CQL when learning from the `noisy` and `biased` AntMaze datasets for different Lagrange threshold value tau. tau=2.0 performs the best overall and is the result we used in Table 1 for CQL.

| Dataset | tau=1.0 | tau=2.0 | tau=0.5 |
|---|---|---|---|
| antmaze-large-bias | 50.0 (5.3) | 61.7 (3.5) | 5.7 (3.1) |
| antmaze-large-noisy | 41.7 (4.6) | 50.3 (2.3) | 5.0 (5.0) |
| antmaze-medium-bias | 72.7 (7.0) | 66.7 (2.9) | 31.7 (13.0) |
| antmaze-medium-noisy | 55.0 (5.3) | 55.7 (4.7) | 17 (3.6) |

**EDAC** To obtain the results for EDAC presented in Table 1, we performed a sweep over the number of value function used in the ensemble and the weight $\eta$ (used in step 5 in Algorithm 1 in EDAC paper to weight the gradient penalty term $ES$). For the number of value function, we ran the sweep over $20, 50, 100$. For the weight $\tau$, we ran the sweep over the values $1, 5, 10, 50, 100, 1000$. In other words, we ran $3 \times 6 = 18$ different combinations of hyper-parameter values for EDAC. We use the implementation of EDAC released by the original authors. Since we discovered that EDAC does not perform well in the medium size maze, we did not obtain the results of EDAC for the large maze.

**BEAR** For BEAR, we used the laplacian kernel and performed a sweep over the hyper-parameter $mmd\_sigma$ using the values $1, 10, 20, 50$. We use the implementation available publicly at `https://github.com/rail-berkeley/d4rl_evaluations`. Similarly to EDAC, because BEAR does not perform well when learning from the medium size maze datasets, we did not obtain the results of BEAR for the large maze.

# F    Detailed algorithm description

Algorithm 1 provides a summary of the training step given a batch of transitions from the offline dataset. In this section, we provide the description of how the different networks in our algorithms are trained using Python syntax. We include four Code Listings below, each illustrating the details of an update step in Algorithm 1.

Code Listing 2: Value networks training step given a batch of data, corresponding to step 4 in Algorithm 1

```python
rewards = batch['rewards']
terminals = batch['terminals']
obs = batch['observations']
actions = batch['actions']
next_obs = batch['next_observations']

# Computing target Q values
next_obs_target_actions = policy(next_obs)

target_Q1 = target_qf1(next_obs, next_obs_target_actions)
target_Q2 = target_qf2(next_obs, next_obs_target_actions)
target_Q = torch.min(target_Q1, target_Q2)
target_Q = rewards + (1 - terminals) * discount * target_Q

# Obtain loss function
current_Q1, current_Q2 = qf1(obs, actions), qf2(obs, actions)

qf1_loss = F.mse_loss(current_Q1, target_Q)
qf2_loss = F.mse_loss(current_Q2, target_Q)

# Update parameters of value functions
qf1_optimizer.zero_grad()
qf1_loss.backward()
qf1_optimizer.step()

qf2_optimizer.zero_grad()
qf2_loss.backward()
qf2_optimizer.step()

# Update Target Networks
soft_update_from_to(qf1, target_qf1, tau)
soft_update_from_to(qf2, target_qf2, tau)
```

Code Listing 3: Policy network training step given a batch of data, corresponding to step 5 in Algorithm 1

```python
obs = batch['observations']
real_actions = batch['actions']

actor_actions = policy(obs)

# Compute value estimate
Q_pi_actions = qf1(obs, actor_actions)

# Compute log probability under discrimator
D_actor_actions_logit = discriminator(
    obs,
    actor_actions,
    return_logit=True
)

log_D_actor_actions = F.logsigmoid(D_actor_actions_logit)

# Compute probability ratio
probs = discriminator(obs, actor_actions)
real_actions_probs = discriminator(obs, real_actions)

probs = torch.min(real_actions_probs, probs)

# min (D(s, a), D(s, a_dataset)) / D(s, a_dataset)
probs = probs / real_actions_probs

probs = probs.detach()

# Compute loss and update policy
policy_loss = - (
    probs * Q_pi_actions / w + log_D_actor_actions
).mean()

policy_optimizer.zero_grad()
policy_loss.backward()
policy_optimizer.step()
```

Code Listing 4: Auxiliary generator training step given a batch of data, corresponding to step 6 in Algorithm 1

```python
obs = batch['observations']

# Calculate loss
b_size = obs.size(0)
real_label = torch.full(
        (b_size,),
        1)

actions_fake = auxiliary_generator(obs)

logits = discriminator(
    obs,
    actions_fake,
    return_logit=True)

err = F.binary_cross_entropy_with_logits(
    logits,
    real_label)

# Update auxiliary generator
auxiliary_generator_optimizer.zero_grad()
err.backward()
auxiliary_generator_optimizer.step()
```

Code Listing 5: Discriminator training step given a batch of data, corresponding to step 7 in Algorithm 1

```python
obs = batch['observations']
actions = batch['actions']

b_size = obs.size(0)

# Calculate loss on real action
D_real_logits = discriminator(
    obs,
    actions + get_instance_noise(actions),
    return_logit=True
)

real_label = torch.full(
        (b_size,),
        1)

errD_real = F.mse_loss(
    F.sigmoid(D_real_logits),
    real_label
) / 2.

# Calculate loss on fake action
def loss_fake_action(fake_action):
    fake_label = torch.full(
        (b_size,),
        0,
    )

    D_fake_logits = discriminator(
        obs,
        fake_action.detach() + get_instance_noise(fake_action),
        return_logit=True
    )

    errD_fake = F.mse_loss(
        F.sigmoid(D_fake_logits),
        fake_label
    ) / 2.

    return errD_fake

fake_action_aux = auxiliary_generator(obs)
fake_action_policy = policy(obs)

err_D_fake = loss_fake_action(fake_action_aux) \
    + loss_fake_action(fake_action_policy)

# Compute gradient and update the discriminator
discriminator_optimizer.zero_grad()
(errD_real + err_D_fake).backward()
discriminator_optimizer.update()
```