# OpenReview forum: "DASCO: Dual-Generator Adversarial Support Constrained Offline Reinforcement Learning"
_NeurIPS.cc/2022/Conference — NeurIPS 2022 Accept_

### Official Review · Reviewer_PvFC · 2022-07-09

**Rating:** 6
**Confidence:** 4
**Soundness:** 3 good
**Presentation:** 3 good
**Contribution:** 3 good

**Summary:**

This paper proposes a GAN-based method to learn an offline policy.

- The policy is trained as the generator of GAN to make the policy close to the supported distribution. Meanwhile, the policy is trained to maximize rewards as an ordinary RL algorithm.
- To handle the potential conflicts in these two objectives, the authors use two separate distributions to learn the policy and the auxiliary generator, respectively.
- The experiments on mixed datasets with sub-optimal data obtain good performance compared to several previous methods.

**Questions:**

1. Can you explain more about "the policy does not need to match the entire data distribution, but only the slice of the data that leads to high long term performance" in the abstract?

2. What is the relationship between Theorem 4.1 and the proposed method? The data distribution weight seems hard to implement. What information can we obtained from the form of the data distribution weight?

3. In Eq.8, why the optimal primary generator  $p^*_g=2P_D$ if $2P_D<1$? What is the "next" best in-support maxima in Eq. 9? Does the probabilities of $p^*_g(x_j)$ are sorted when you sum $j=0$ to $i$? Can you give an example to explain what distribution of $p^*_g(x_j)$ should be in simple cases?

4. The ablation study shows dynamic weighting in Eq. 11 is important. It seems that the policy has be constrained to be close to the bebavior policy through adversarial training, why the dynamic weighting is still important especially in antmaze tasks?

5. The relationship between this paper and uncertainty-based methods should be discussed, like EDAC (NeurIPS 21) and PBRL (ICLR 22).


**Ethics Review Area:**

["I don’t know"]

**Strengths And Weaknesses:**

Strengths
- The idea that uses GAN to model the state-action distribution of offline dataset is novel. Compared to directly learn a behavior policy, the GAN-based method is more flexible and enables the network to learn complex distribution from a mixed dataset.
- The policy constraints is applied implicitly by using the policy network as the generator of GAN.

Weaknesses
- Some conclusions and theoretical analysis need more explanation (see questions)
- Several related works should be discussed.
- Since the method applies adversarial training that is usually considered unstable, it is better to release the public code to ensure reproducibility.

---

> ### Author Response · Authors · 2022-08-02
> **Reply to reviewer on August 2 (Part 1 of 2)**
>
> We thank the reviewer for their comments!
>
> **We added Section B in the Appendix**, which contains additional experiments to understand the role of the auxiliary generator in a simple toy task.
>
> We provide responses to individual questions raised in the review below. Please let us know if these responses address the most significant concerns raised in your review, or if any other issues remain!
>
> ___
>
> **Can you give an example to explain what distribution of $ p^\*_{g} (x_j) $ should be in simple cases?**
>
> We added an one-dimensional numerical example in Section B in the Appendix. The example illustrates the difference of the coverage/distribution with and without the auxiliary generator and demonstrates the benefit of the dual generator technique, both visually and numerically.
>
> ___
>
> **What is the relationship between Theorem 4.1 and the proposed method? The data distribution weight seems hard to implement. What information can we obtained from the form of the data distribution weight?**
>
> We interpret Theorem 4.1 to mean that using a single generator is **fundamentally** limited, because maximizing the discriminator score and the value function can be conflicting objectives. Therefore, the learned generator can fail to maximize either objective.
>
> As such, Theorem 4.1 serves two functions:
>
> 1. Provide insights as to why previous GAN-based offline RL methods have failed to perform as well as alternative methods
>
> 2. Suggest the solution, which is our proposed dual generator technique
>
> These two points are discussed in the paper at the end of page 4.
>
> **The data distribution weight seems hard to implement.** => We do not need to implement or compute the data distribution weight in our algorithm. The data distribution weight only appears in the theoretical analyses of the algorithm.
>
> **What information can we obtained from the form of the data distribution weight?** => From Theorem 4.1, we can see that the optimal generator does not attempt to match the data distribution anymore. As such, when the data distribution weight $ \dfrac{ e^{- f(x) - \nu } }{ 2 - e^{ - f(x) - \nu } } $ brings the optimal generator solution far away from the data distribution, the discriminator clearly has an advantage in learning to distinguish between real samples and sample generated by the generator. In this case, we would expect a collapse in learning performance.
>
> ___
>
> **In Eq.8, why the optimal primary generator $ p^\*_{g} = 2 p_D $ if $ 2 p_D < 1 $ ?**
>
> This is because:
>
> The value $ x_0 $ in Eq. 8 is the global optima of the secondary objective function $ f $. Therefore, the **optimal** primary generator $ p^\*_{g} $ should assign the highest possible probability to the value $ x_0 $.
>
> However, we have Eq. 7, which we restate below
>
> $ p^*_{aux} = 2p_D - p^*_{g} $
>
> In this equation, the LHS is a probability, which must be larger than or equal to 0. As such, we have the constraint that $ 2p_D - p^*_{g} \geq 0 $, which implies  $ 2p_D \geq p^*_{g} $. As such, for any value $ x $, the optimal primary generator can assign at most $ 2p_D ( x ) $ probability to $ x $.
>
> In other words, for $ x_0 $, the **optimal** primary generator $ p^*_{g} $ should assign the highest possible probability to the value $ x_0 $, and the highest probability  $ p^*_{g} $ can assign to $ x_0 $ is $ 2p_D ( x ) $.
>
> Therefore, $ p^*_{g} (x_0) = 2 p_D (x_0) $ if $ 2 p_D (x_0) < 1 $.
>
> We have added more explanation to develop intuition for the solution on page 5 in the revised paper.
>
> ___
>
> **What is the "next" best in-support maxima in Eq. 9?**
>
> We note that it should be “minima” instead of “maxima”. Our apologies for the typo! We provide the answer to your question below:
>
> Given $ x_0 $ is the global in-support optimum of $ f $, then $ x_1 $ is the **next** best in-support optimum of $ f $ in the sense that:
>
> 1. The behavior policy assigns non-zero probability to $ x_1 $. That is, $ p_D(x_1) > 0 $. (condition 1)
>
> 2. $ x_1 $ is not better than $ x_0 $ at minimizing $ f $, that is, $ f(x_0) \leq f(x_1)  $. (condition 2)
>
> 3. Other than $ x_0 $, there does not exist another value for $ x $ that is both in-support (condition 1) and is better than $ x_1 $ at minimizing $ f $. (condition 2)
>
> Given $x_0$ and $x_1$ defined as above, we can then **similarly** define $x_2$ to be the **next** best in-support optimum if:
>
> 1. The behavior policy assigns non-zero probability to $ x_2 $.
>
> 2. $ x_2 $ is not better than $ x_0 $ or $ x_1 $ at minimizing $ f $, that is, $ f(x_0) \leq f(x_2)  $ and  $ f(x_1) \leq f(x_2) $
>
> Using these two conditions, given the first $i-1$ in-support optimum of $ f $, we can define the **next** best in-support optimum $ x_i $, as we have done in Eq 9.

---

> > ### Author Response · Authors · 2022-08-02
> > **Reply to reviewer on August 2 (Part 2 of 2)**
> >
> > **Does the probabilities of  $ p^\*_{g} (x_j) $ are sorted when you sum $ j = 0 $ to $ 1 $ ?**
> >
> > Does the reviewer mean $ i $ instead of $ 1 $ at the end of your question?
> >
> > Sorted probabilities are actually not a requirement of our theoretical analyses. Our theoretical analyses do require that the $ x_j $ where $ j $ goes from $ 0 $ to $ i $ are sorted by the value assigned by the secondary function $ f $ in ascending order.
> >
> > However, please note that this condition is only required in the theoretical analyses of the form of the optimal solution for the primary generator. We do **not** need this condition when implementing the actual computational algorithm. In other words, the sorted order of the $ x_j $ is induced by the use of the dual generator technique.
> >
> > ___
> >
> > **The ablation study shows dynamic weighting in Eq. 11 is important. It seems that the policy has be constrained to be close to the bebavior policy through adversarial training, why the dynamic weighting is still important especially in antmaze tasks?**
> >
> > To clarify, the policy is not constrained to be close to the behavior policy through adversarial training, which would be a distribution constraint. Our algorithm enforces support constraint, which only requires that the policy action is not too unlikely under the behavior policy.
> >
> > Regarding why we still need the dynamic weight, the constraint is enforced **in expectation**. In other words, even when the policy is close to the behavior policy in expectation, the policy can still output action that is unlikely under the behavior policy. As we explain in Section 4.2, when such policy action is sampled during the policy improvement step, the sampled actions will exploit spurious maxima in the value function and have their probability of being sampled again in the future increased. When the same actions are sampled during the policy evaluation step, the errors in the value functions from the next states are backed up into the preceding states, leading to divergent value functions.
> >
> > As to why dynamic weighting is more important in AntMaze tasks, we hypothesize this is because of the property of the offline dataset and the task. In AntMaze, the offline dataset can contain no single trajectory that witnesses the Ant successfully navigating from the start to goal location. As such, the learning algorithm needs to stitch together sub-trajectories to successfully accomplish the task. On the other hand, the HalfCheetah and Hopper datasets contain a fairly large number of successful trajectories. Intuitively, stitching together sub-trajectories requires more policy improvement steps, and hence more room for the policy to exploit spurious maxima of the value function.
> >
> > ___
> >
> > **The relationship between this paper and uncertainty-based methods should be discussed, like EDAC (NeurIPS 21) and PBRL (ICLR 22).**
> >
> > Thank you for mentioning these two references! We have added a discussion of these two papers to the related work section.
> >
> > We also demonstrated that EDAC has poor performance on the AntMaze tasks and included the result in Table 1 in the experiment section.
> >
> > ___
> >
> > **Since the method applies adversarial training that is usually considered unstable, it is better to release the public code to ensure reproducibility.**
> >
> > We agree that the code should be released to ensure reproducibility. We are working on removing proprietary tooling from the codebase, and will try to release the code before the Author - Reviewer discussion period ends.
> >
> > ___
> >
> > **Can you explain more about "the policy does not need to match the entire data distribution, but only the slice of the data that leads to high long term performance" in the abstract?**
> >
> > This sentence is referring to the difference between **distribution** and **support** constraint. In distribution constraint, the policy is required to be close to the behavior policy. This is not preferable when the behavior policy has poor performance or is random. In contrast, support constraint only requires the policy to output actions that are not too unlikely under the behavior policy. The importance of support constraint has also been echoed by other researchers in offline RL, such as in [1].
> >
> > In other words, if there are actions that lead to poor performance and are very likely under the behavior policy, support constraint does **not** require the learned policy to output these actions.
> >
> > Please let us know if our explanation is clear and we are happy to follow up.
> >
> > [1] https://bair.berkeley.edu/blog/2019/12/05/bear/

---

> > > ### Author Response · Authors · 2022-08-04
> > > **Dear Reviewer,**
> > >
> > > When you get the chance, can you please kindly have a look at our reply and let us know if your concerns have been addressed, or if you would like additional follow-up. Thank you!

---

> > > > ### Author Response · Authors · 2022-08-07
> > > > **Dear Reviewer**
> > > >
> > > > Can you please take a look at our reply and let us know if you still have concerns with the paper when you get the chance?
> > > >
> > > > There is only 3 days left in the author - reviewer discussion period, and we would love to hear if you are happy with our reply to your comments.
> > > >
> > > > Thank you!

---

> > > > > ### Comment · Reviewer_PvFC · 2022-08-07
> > > > > **Further questions**
> > > > >
> > > > > 1. "if there are actions that lead to poor performance and are very likely under the behavior policy, support constraint does not require the learned policy to output these actions." How to achieve this?
> > > > >
> > > > > 2. I wonder the three situations in Eq. 9 are only for analysis, or have some guidance in practical implementation and experiments?

---

> > > > > > ### Author Response · Authors · 2022-08-07
> > > > > > **Reply to reviewer on August 7**
> > > > > >
> > > > > > **1. "if there are actions that lead to poor performance and are very likely under the behavior policy, support constraint does not require the learned policy to output these actions." How to achieve this?**
> > > > > >
> > > > > > Achieving this goal is precisely the main contribution of the dual generator technique. For the actions that lead to **poor performance** and are very **likely under the behavior policy**, the auxiliary generator (not the learned policy) will learn to output these actions. As such, thanks to the supporting role that the auxiliary generator plays, the primary generator (or equivalently the learned policy) is free to only output the actions that lead to good performance.
> > > > > >
> > > > > > In other words, using the dual generator technique induces the above property of the learned policy.
> > > > > >
> > > > > > ---
> > > > > >
> > > > > > **2. I wonder the three situations in Eq. 9 are only for analysis, or have some guidance in practical implementation and experiments?**
> > > > > >
> > > > > > Thank you for this question!
> > > > > >
> > > > > > The three situations in Eq. 9 can indeed be used in practical implementation. For instance, in the example we provide in Appendix B, we are able to find the optimal solution for the primary generator **in closed form** thanks to Eq. 9.
> > > > > >
> > > > > > Please don't hesitate to let us know if you have more questions or would like additional clarifications.
> > > > > >
> > > > > > Thank you very much!

---

> > > > > > > ### Author Response · Authors · 2022-08-08
> > > > > > > **Did we answer your question?**
> > > > > > >
> > > > > > > Dear Reviewer,
> > > > > > >
> > > > > > > Since there are only 24 hours left in the discussion period, we would love to hear if you have any remaining questions or concerns for us!
> > > > > > >
> > > > > > > Thank you very much!

---

### Official Review · Reviewer_nQaE · 2022-07-11

**Rating:** 7
**Confidence:** 4
**Soundness:** 3 good
**Presentation:** 3 good
**Contribution:** 3 good

**Summary:**

This paper proposes a novel GAN-based approach to tackle the distributional shift issue in offline RL. The authors noticed that the conventional single generation GAN framework requires the generator to both maximize the expected return and match the dataset distribution, which is challenging and allows the discriminator dominate the generator. To address this problem, the authors proposed to add an auxiliary generator to aid the distributional matching. The main generator will be trained to both match the data distribution and maximize the expected return while the auxiliary generator focuses on learning the "remainder" of the data distribution that the main generator does not learn well. The authors did extensive analysis and experiments with good ablation studies to demonstrate the effectiveness of the proposed method.

**Questions:**

Have the authors encountered challenges when training the dual generator GAN? Are any special tricks required to stabilize the GAN training?


**Limitations:**

The authors didn't provide a discussion on the limitations and potential negative societal impact of their work. Please mention the offline evaluation challenge to allow the audience understand that the offline learned RL policy cannot be easily deployed.

**Strengths And Weaknesses:**

### Strengths
1. The proposed dual-generator is well motivated and overall writing is easy to follow.
2. The proposed method outperforms strong offline RL baselines based on distribution constraints.
3. The ablation study provides a nice way to understand the effect of the auxiliary generator.

### Weakness
1. The experiments are performed on relatively simple navigation and locomotion tasks. It would be nice to see results on more complex manipulation tasks.

---

> ### Author Response · Authors · 2022-08-02
> **Reply to reviewer on August 2**
>
> We thank the reviewer for their comments!
>
> In the Conclusion section, we added a standalone discussion on potential negative societal impacts.
>
> Below, we provide a more detailed reply to your comments. Please let us know if these responses address the most significant concerns raised in your review, or if any other issues remain!
>
> ___
>
> **Have the authors encountered challenges when training the dual generator GAN? Are any special tricks required to stabilize the GAN training?**
>
> We found that fairly standard methods for stabilizing GANs were very effective for our method. The particular stabilization techniques we used are described in Appendix C, and include:
>
> 1. Using Least Square GAN objective function to provide smooth and non-saturating gradient in the discriminator [1]
>
> 2. Adding noise to the action before inputting the action to the discriminator, the so called “instance noise” technique [2]
>
> 3. Training the discriminator for more steps than the generators in each algorithm step (5 training steps for the discriminator for every step of the generator)
>
> [1] https://openaccess.thecvf.com/content_ICCV_2017/papers/Mao_Least_Squares_Generative_ICCV_2017_paper.pdf
>
> [2] https://www.inference.vc/instance-noise-a-trick-for-stabilising-gan-training/
>
> ___
>
> **The authors didn't provide a discussion on the limitations and potential negative societal impact of their work. Please mention the offline evaluation challenge to allow the audience understand that the offline learned RL policy cannot be easily deployed.**
>
> Thank you for bringing this up! In our submission, we briefly discussed the potential societal impact of the work when answering question 1(c) in the checklist on page 12.
>
> We have added a standalone discussion on the potential societal impact of the work at the end of the Conclusion section in the main paper. We also included a discussion regarding offline evaluation and the challenges related to deploying offline learned RL policy on real world problems.
>
> ___
>
> **The experiments are performed on relatively simple navigation and locomotion tasks. It would be nice to see results on more complex manipulation tasks.**
>
> We agree with the reviewer that testing the algorithms on more complex tasks would have made the case for our paper stronger. Nevertheless, we believe that our current experimental setting has the essence of a class of hard problems that require combination and stitching of various sub-optimal experiences. The AntMaze setting allows us to investigate the performance of offline RL algorithms in scenarios where support constraints are important, without other confounding challenges, such as learning from pixel input.
>
> As demonstrated in Table 1, current algorithms struggle with this setting, whereas our method can learn high-performing behaviors.

---

> > ### Author Response · Authors · 2022-08-04
> > **Dear Reviewer,**
> >
> > When you get the chance, can you please kindly have a look at our reply and let us know if your concerns have been addressed, or if you would like additional follow-up. Thank you!

---

> > > ### Author Response · Authors · 2022-08-07
> > > **Dear Reviewer**
> > >
> > > Can you please take a look at our reply and let us know if you still have concerns with the paper when you get the chance?
> > >
> > > There is only 3 days left in the author - reviewer discussion period, and we would love to hear if you are happy with our reply to your comments.
> > >
> > > Thank you!

---

> > > > ### Comment · Reviewer_nQaE · 2022-08-07
> > > > **.**
> > > >
> > > > Thanks for responding to my comments. I will keep my original rating of 7.

---

> > > > > ### Author Response · Authors · 2022-08-07
> > > > > **Thank you for getting back to us!**
> > > > >
> > > > > We would love to hear if you have any remaining concerns or comments for the paper.

---

> ### Comment · Area_Chair_3gDW · 2022-08-07
> **Discussion with authors**
>
> Could you please acknowledge that you have read the response to your review? Also, please reply to the authors to indicate how they managed to answer the points raised in your review and how this impacts your score. Finally, make sure that you update your score accordingly.

---

### Official Review · Reviewer_uYzh · 2022-07-12

**Rating:** 5
**Confidence:** 4
**Soundness:** 3 good
**Presentation:** 2 fair
**Contribution:** 2 fair

**Summary:**

This paper proposed to resolve the conflict issue when incorporating a generative adversarial network (GAN) in an offline RL setting. The authors argue that optimizing the action-value function and the discriminator during policy improvement causes poor performance on either objective. The authors showed that using an auxiliary generator alleviates the conflict issue, and RL policy attains in-support optimization of the Q-value function. Additional importance weight on the Q-value further reduced the possibility of extrapolation errors. Numerical results show that the proposed method produced competitive performance in the standard benchmark tasks and the best performance in the noisy/biased datasets.



**Questions:**

1. GAN has a stability issue. Training of both generator and discriminator requires careful design (architecture, update period, and so on). I suspect that inserting an additional generator still causes the stability problem. Did the proposed method suffer from the stability issue? How did the authors handle this issue? (Importance weight helps alleviate extrapolation error regarding RL agent, not explicitly guaranteeing the stability of GAN). What are the update periods of D, G, and the auxiliary generator?

2. To improve the quality of the paper, it is a good idea to show the visual difference of the coverage/distribution with and without the auxiliary generator to support the claimed in-support optimization.

3. It looks like there is a gap converting $\min_\pi \log(1-D(s,a))$ in Eq 2 and 13 to $\max_\pi \log D(s,a)$ without any explanation. In fact, this is not supported by the claimed mathematical proofs. Did the authors also conducted experiment using $\min_\pi \log(1-D(s,a))$ in Eq 13?

4. Is there any theoretical insight of adopting importance weight in Eq 11?

5. Why did the authors use only CQL and IQL for the baselines in Tables 1 and 2?

6.  Did the authors tune key parameters in CQL and IQL for the noisy/biased dataset? (e.g., \tau in IQL) If yes, what are the values?

7. Why were the model-based offline RL methods not considered in Table 3?

8. For clarity, does 'without (auxiliary generator)' in Table 4 use importance weight?


**Limitations:**

The authors did not address a potential societal impact of their work.



**Strengths And Weaknesses:**

Pros
  1) The authors tried to explain the proposed method with mathematical justifications.
  2) The authors provide a reasonable ablation study.

Cons
  1) Several parts of the paper should be clarified. (Please see the below Questions.)
  2) Additional experiments are required. (Please see the below Questions.)

---

> ### Author Response · Authors · 2022-08-02
> **Reply to reviewer on August 2 (Part 1 of 2)**
>
> Thank you for your comments!
>
> **We added Section B in the Appendix**, which contains additional experiments to understand the role of the auxiliary generator in a simple toy task.
>
> We provide responses to individual questions raised in the review below. Please let us know if these responses address the most significant concerns raised in your review, or if any other issues remain!
>
> ___
>
> **To improve the quality of the paper, it is a good idea to show the visual difference of the coverage/distribution with and without the auxiliary generator to support the claimed in-support optimization.**
>
> Thank you for your suggestion!
>
> We added an one-dimensional numerical example in Section B in the Appendix. The example illustrates the difference of the coverage/distribution with and without the auxiliary generator and demonstrates the benefit of the dual generator technique, both visually and numerically.
>
> We also highlight our ablation study presented in Table 5. In this table, we compare the performance of the primary generator and the auxiliary generator. Given that the two generators are trained using the same dataset, the difference in their performance implies that they have learned to output different subsets of the data distribution.
>
> ___
>
> **Why did the authors use only CQL and IQL for the baselines in Tables 1 and 2?**
>
> In our original submission, we use CQL and IQL as baselines because these are recent state of the art model-free algorithms, that seem to be most closely related to our algorithm.
>
> We have also added two more baselines, BEAR and EDAC, to Table 1, and find that DASCO still outperforms these prior works.
>
> ___
>
> **Did the authors tune key parameters in CQL and IQL for the noisy/biased dataset? (e.g., \tau in IQL) If yes, what are the values?**
>
> For IQL, we tried 5 different values of the expectile parameters tau, which are 1.0, 0.95, 0.9, 0.8, 0.7. For CQL we tuned the Lagrange threshold over three values (2.0, 1.0, 0.5). For both algorithms, their best performance after tuning is still worse than our method, as shown in Table 1.
>
> To provide more details on how the baselines are tuned when learning from the noisy/biased dataset, we also added Section E in the Appendix.
>
> ___
>
> **Is there any theoretical insight of adopting importance weight in Eq 11?**
>
> Adopting the importance weight in Eq 11 follows from our observation that the discriminator scores can serve as an effective way to avoid the erroneous overestimation in the Q-function from driving the policy towards a suboptimal action. In Eq 11, we multiply the state-action values with the discriminator scores in the policy objective. In doing so, the more unlikely an action is under the behavior policy, the more the Q-value of this action is down-weighted when we update the policy
>
> ___
>
> **GAN has a stability issue. Training of both generator and discriminator requires careful design (architecture, update period, and so on). I suspect that inserting an additional generator still causes the stability problem. Did the proposed method suffer from the stability issue? How did the authors handle this issue? (Importance weight helps alleviate extrapolation error regarding RL agent, not explicitly guaranteeing the stability of GAN). What are the update periods of D, G, and the auxiliary generator?**
>
> While we did observe challenges with stability early on in the project, we found that fairly standard methods for stabilizing GANs were very effective for our method, and alleviated these challenges. The particular stabilization techniques we used are described in Appendix C, and include:
>
> 1. Using Least Square GAN objective function to provide smooth and non-saturating gradient in the discriminator [1]
>
> 2. Adding noise to the action before inputting the action to the discriminator, the so called “instance noise” technique [2]
>
> 3. Training the discriminator for more steps than the generators in each algorithm step (5 training steps for the discriminator for every step of the generator)
>
> [1] https://openaccess.thecvf.com/content_ICCV_2017/papers/Mao_Least_Squares_Generative_ICCV_2017_paper.pdf
>
> [2] https://www.inference.vc/instance-noise-a-trick-for-stabilising-gan-training/

---

> > ### Author Response · Authors · 2022-08-02
> > **Reply to reviewer on August 2 (Part 2 of 2)**
> >
> > **It looks like there is a gap converting $ min_\pi \log ( 1 - D(s, a) ) $ in Eq 2 and 13 to $max_\pi \log D(s, a) $ without any explanation. In fact, this is not supported by the claimed mathematical proofs. Did the authors also conducted experiment using $ min_\pi \log ( 1 - D(s, a) ) $ in Eq 13?**
> >
> > We use $ max_\pi \log D(s, a) $ instead of $ min_\pi \log ( 1 - D(s, a) )$ because the original GAN paper recommends training the generator to maximize $ \log D(s, a) $ instead of to minimize $ \log ( 1 - D(s, a) ) $ to avoid vanishing gradient. For reference, please see the third paragraph on page 3 of the original GAN paper [3].
> >
> > Using $ max_\pi \log D(s, a) $ instead of $ min_\pi \log ( 1 - D(s, a) ) $ has also become a standard technique in training GAN. Please refer to slide 111 to 113 in this set of slides for example [4].
> >
> > [3] https://arxiv.org/pdf/1406.2661.pdf
> >
> > [4] http://cs231n.stanford.edu/slides/2019/cs231n_2019_lecture11.pdf
> >
> > ___
> >
> > **Why were the model-based offline RL methods not considered in Table 3?**
> >
> > For table 3, we took the performance of the baselines from Table 1 in the IQL paper, which does not include model-based offline RL methods. However, we have also added a model-based algorithm (referred to as COMBO) to the result in Table 3.
> >
> > ___
> >
> > **For clarity, does 'without (auxiliary generator)' in Table 4 use importance weight?**
> >
> > Yes it does.
> >
> > ___
> >
> > **The authors did not address a potential societal impact of their work.**
> >
> > Thank you for bringing this up! In our submission, we briefly discussed the potential societal impact of the work when answering question 1(c) in the checklist on page 12.
> >
> > We have added a standalone discussion on the potential societal impact of the work at the end of the Conclusion section in the main paper.

---

> > > ### Author Response · Authors · 2022-08-04
> > > **Dear Reviewer,**
> > >
> > > When you get the chance, can you please kindly have a look at our reply and let us know if your concerns have been addressed, or if you would like additional follow-up. Thank you!

---

> > > > ### Author Response · Authors · 2022-08-07
> > > > **Dear Reviewer**
> > > >
> > > > Can you please take a look at our reply and let us know if you still have concerns with the paper when you get the chance?
> > > >
> > > > There is only 3 days left in the author - reviewer discussion period, and we would love to hear if you are happy with our reply to your comments.
> > > >
> > > > Thank you!

---

> > > > > ### Comment · Reviewer_uYzh · 2022-08-07
> > > > > **Dear Authors**
> > > > >
> > > > > Thank you for the reply by the authors. I read the reply carefully and checked the additional results in the revised draft.

---

> > > > > > ### Author Response · Authors · 2022-08-08
> > > > > > **Thank you for getting back to us!**
> > > > > >
> > > > > > We would love to hear if you have more questions or comments for the paper!

---

> ### Comment · Area_Chair_3gDW · 2022-08-07
> **Discussion with authors**
>
> Thank you for acknowledging that you have read the response to your review.
>
> Could you please reply to the authors to indicate how they managed to answer the points raised in your review and how this impacts your score? Finally, make sure that you update your score accordingly.

---

### Official Review · Reviewer_kSDN · 2022-07-12

**Rating:** 6
**Confidence:** 2
**Soundness:** 3 good
**Presentation:** 3 good
**Contribution:** 3 good

**Summary:**

This paper proposes a modified GAN-based policy learning objective that modifies existing work. The use of a second (auxiliary) generator to capture additional objective is quite interesting. The algorithm outperforms existing offline RL methods in noisy and biased dataset that resemble robotics control and data relabeling. There are some problems with writing on the experiment section of the paper and additional experiments need to be run. If these problems can be fixed, it will be a good paper meeting the acceptance criteria.

I'm happy to update my score if my questions are addressed and/or if I missed several key factors/findings of the paper.

**Questions:**

- Can Table 1 include more baselines, beyond CQL and AWR (or IQL)? Or is DASCO performing worse or on-par if you include other baselines?
- Can the authors explain how to sample from $p_{mix}$?
- Can authors explain a bit more on why does the biased dataset resemble data relabeling? In your case, you are adding bias/noise to state and action. However, normal transition relabeling is to relabel reward instead.

**Limitations:**

The paper lacks experimental evidence that convinces me this algorithm can solve a unique challenge that other algorithms can't (even though the introduction/conclusion makes me believe that there should be such use case).

**Strengths And Weaknesses:**

Strengths
- The first half of the paper is very well written.
- I quickly read through the theorem and the proof. The approach generally seems principled, but I'm unable to comment on how "innovative" this modification is and whether it is an incremental improvement from the previous GAN-based offline policy RL objective.
- The biased and noisy datasets are designed with specific scenarios in mind. DASCO does outperform two other baselines with a large margin.

Weakness
- The second half of the paper is a bit harder to interpret.
  - Starting Section 5 (Experiments), some details of the experimental setup gets lost. For example, baseline 10% BC is not explained neither in main text nor appendix (but please correct me if I missed it).
  - Line 226-227 says "Advantage Weighted Regression [31] (IQL)", first of all, it should be AWR, not IQL (a different paper). The citation suggests AWR as well. So is the experiment done with IQL or AWR?
- Lack of better use case
  - Proposing new algorithms will always have its value -- we shouldn't force every new algorithm to be "SOTA" in order to get published. The idea discussed in this paper seems interesting enough that I don't think I'm too hang up on the fact that Table 2 and Table 3 shows close performance with other baselines. However, I am a believer that there should be some unique use cases that this algorithm can and will be used by others. Table 1 seems to be getting this point across, but 2 tasks / settings are very limited and it's only compared against 2 baselines, therefore not very trustworthy.

---

> ### Author Response · Authors · 2022-08-02
> **Reply to reviewer on August 2 (Part 1 of 2)**
>
> Thank you for your comments!
>
> **We added Section B in the Appendix**, which contains additional experiments to understand the role of the auxiliary generator in a simple toy task.
>
> We provide responses to individual questions raised in the review below. Please let us know if these responses address the most significant concerns raised in your review, or if any other issues remain!
>
> ___
>
> **Lack of better use case
> Proposing new algorithms will always have its value -- we shouldn't force every new algorithm to be "SOTA" in order to get published. The idea discussed in this paper seems interesting enough that I don't think I'm too hang up on the fact that Table 2 and Table 3 shows close performance with other baselines. However, I am a believer that there should be some unique use cases that this algorithm can and will be used by others. Table 1 seems to be getting this point across, but 2 tasks / settings are very limited and it's only compared against 2 baselines, therefore not very trustworthy.**
>
> To address your concerns about Table 1, we added additional baselines (discussed below) to each of the four experiments.
>
> Tables 2 and 3 show comparisons on the standard set of D4RL benchmark tasks. These tasks are used in numerous offline RL comparisons, and while the improvement (in terms of average overall score) on these tasks is not huge, the increment over prior works is well in line with improvements attained in other published papers (see, e.g., IQL, CQL versus Onestep RL, TD3+BC).
>
> ___
>
> **Can Table 1 include more baselines, beyond CQL and AWR (or IQL)? Or is DASCO performing worse or on-par if you include other baselines?**
>
> We added two more recent algorithms, BEAR [1] and EDAC [2], to Table 1. However, compared to CQL, IQL and our methods, BEAR and EDAC have significantly poorer performance.
>
> We chose BEAR because the BEAR paper also argues for support constraint over distribution constraint when designing new offline RL algorithms.
>
> We chose EDAC because the method is relatively recent (NeurIPS 2021) and also maintain a measure of uncertainty. In our method, we use the discriminator score as the measure of uncertainty.
>
> [1] https://arxiv.org/abs/1906.00949
>
> [2] https://arxiv.org/abs/2110.01548
>
> ___
>
> **I quickly read through the theorem and the proof. The approach generally seems principled, but I'm unable to comment on how "innovative" this modification is and whether it is an incremental improvement from the previous GAN-based offline policy RL objective.**
>
> We think perhaps there is a misunderstanding and would like to clarify that our method does not use a standard GAN. The main contribution is the **novel** auxiliary generator technique. We discuss the theoretical motivation for the technique in Section 4.1. Our experiments in Section 5 demonstrates that the technique can lead to significant performance improvement.
>
> ___
>
> **Can authors explain a bit more on why does the biased dataset resemble data relabeling?
> In your case, you are adding bias/noise to state and action. However, normal transition relabeling is to relabel reward instead.**
>
> To clarify, we only add bias and noise to the action, and not the state. When generating the offline datasets, if we add bias and noise to the action, the next state is obtained using the ground truth dynamics function of the task.
>
> We explain the intuition using a toy example. Given two datasets A and B, collected from learning to navigate to two different locations A and B. If we relabel the transitions in dataset A with the reward function from task B, then the actions from dataset A would likely exhibit systematic bias compared to actions in dataset B. The systematic bias is that actions from dataset A would bring the agent closer to goal location A, instead of goal location B.
>
> ___
>
> **Starting Section 5 (Experiments), some details of the experimental setup gets lost. For example, baseline 10% BC is not explained neither in main text nor appendix (but please correct me if I missed it).**
>
> 10%BC is referring to performing BC using the top 10% trajectories with highest episode return. We added this clarification to page 8 in the revised paper. Thank you for your suggestion!
>
> We have also added more experimental details, such as how the baselines are tuned when learning from the noisy/biased AntMaze dataset (Appendix E) and citations for the baselines in Table 3.
>
> ___
>
> **Line 226-227 says "Advantage Weighted Regression [31] (IQL)", first of all, it should be AWR, not IQL (a different paper). The citation suggests AWR as well. So is the experiment done with IQL or AWR?**
>
> The experiment is done with IQL. Sorry about the confusion! We mentioned AWR because IQL uses AWR to extract the policy from the learned Q function (This is discussed in section 4.3 in the IQL paper)

---

> > ### Author Response · Authors · 2022-08-02
> > **Reply to reviewer on August 2 (Part 2 of 2)**
> >
> > **Can the authors explain how to sample from p_mix ?**
> >
> > Sampling from p_mix is the same as sampling from a mixture distribution. We randomly choose either the auxiliary generator or the primary generator, because the mixing weight is $\dfrac{1}{2}$. We then sample one action from the chosen distribution. The sampled action is a sample from p_mix.
> >
> > ___
> >
> > **The paper lacks experimental evidence that convinces me this algorithm can solve a unique challenge that other algorithms can't (even though the introduction/conclusion makes me believe that there should be such use case).**
> >
> > We agree with the reviewer that testing the algorithms on more complex tasks would have made the case for our paper stronger. Nevertheless, we believe that our current experimental setting has the essence of a class of hard problems that require combination and stitching of various sub-optimal experiences. The AntMaze setting allows us to investigate the performance of offline RL algorithms in scenarios where support constraints are important, without other confounding challenges, such as learning from pixel input.
> >
> > As demonstrated in Table 1, current algorithms struggle with this setting, whereas our method can learn high-performing behaviors.

---

> > > ### Author Response · Authors · 2022-08-04
> > > **Dear Reviewer,**
> > >
> > > When you get the chance, can you please kindly have a look at our reply and let us know if your concerns have been addressed, or if you would like additional follow-up. Thank you!

---

> > > > ### Author Response · Authors · 2022-08-07
> > > > **Dear Reviewer**
> > > >
> > > > Can you please take a look at our reply and let us know if you still have concerns with the paper when you get the chance?
> > > >
> > > > There is only 3 days left in the author - reviewer discussion period, and we would love to hear if you are happy with our reply to your comments.
> > > >
> > > > Thank you!

---

> > > > > ### Comment · Reviewer_kSDN · 2022-08-07
> > > > > **Dear Authors**
> > > > >
> > > > > I have read your responses. I will increase the score to 6. Thanks for providing detailed explanations to all my questions.

---

> > > > > > ### Author Response · Authors · 2022-08-07
> > > > > > **Thank you so much for getting back to us!**
> > > > > >
> > > > > > We would love to hear if you have any remaining concerns or comments for us!

---

> ### Comment · Area_Chair_3gDW · 2022-08-07
> **Discussion with authors**
>
> Could you please acknowledge that you have read the response to your review? Also, please reply to the authors to indicate how they managed to answer the points raised in your review and how this impacts your score. Finally, make sure that you update your score accordingly.

---

### Official Review · Reviewer_Y2tj · 2022-07-24

**Rating:** 6
**Confidence:** 2
**Soundness:** 3 good
**Presentation:** 3 good
**Contribution:** 3 good

**Summary:**

This paper proposes a new method for performing offline reinforcement learning in the face of low-quality data via modelling two separate generator networks - one capturing the policy and maximizing return, and the other auxiliary network capturing the adjustments needed to fool the discriminator network - and together with a value function weight, they empirically show that their method surpasses previous methods in the setting of noisy, biased data.

**Questions:**

* What is the effect of adding only bias in the Antmaze setting?
* What do the poor actions in the auxiliary network look like for the AntMaze without the policy network? I am surprised it performed so poorly given that the performance in other domains seem decent.
* Are there settings for the bias/noisy data in the real world which are analogous where a similar biasing/noising effect plays out?

**Limitations:**

By far the greatest improvements lie in the setting of the bias, noisy data, but the setting is unfortunately quite artificial with hardcoded biases and noises and producing results in a setting which naturally induces biases/noise in the action would be more convincing.

**Strengths And Weaknesses:**

+ Theoretical insight as to why prior work in modelling with GANs fail in Theorem 4.1
+ Simple yet theoretically convincing modifications to GANs to enable the training to perform well
+ Thorough ablations done with evidence on the boost in performance due to the auxiliary generator network and value function weighting
- The experimental setting of the adding noise and bias to the AntMaze seems artificial, which is unfortunate because it underlies the primary result of the paper
- Standard deviation reported only for a portion of the results in the Appendix
- Some awkward phrasing made the paper hard to read at points

---

> ### Author Response · Authors · 2022-08-02
> **Reply to reviewer on August 2**
>
> Thank you for your detailed feedback!
>
> **We added the standard deviation** of performance to the AntMaze tasks (Table 1, 2) and the ablations (Section 5.2).
>
> In the process of responding to the other reviewers, we also **obtained the results of two more algorithms on the AntMaze domains**. Interestingly, we found them to have poor performance in the AntMaze domains while performance in other domains are decent, making our results on AntMaze less surprising.
>
> We provide responses to individual questions raised in the review below. Please let us know if these responses address the most significant concerns raised in your review, or if any other issues remain!
>
> ___
>
> **Some awkward phrasing made the paper hard to read at points**
>
> We edited the paper to improve readability, including an explanation of the intuition behind Theorem 4.2 in Section 4.1.
>
> We would love to learn more about where the awkward phrasings exist in the paper, and offer clarifications and rewrite the paper in these locations.
>
> ___
>
> **What do the poor actions in the auxiliary network look like for the AntMaze without the policy network? I am surprised it performed so poorly given that the performance in other domains seem decent.**
>
> It would be great if the reviewer can clarify if this comment refers to the results in Table 5?
>
> Qualitatively, we noticed that there are two main reasons for the low performance of the auxiliary network in Table 5: the ant either fails to navigate to the goal location, or it would flip on its back, thereby rendering further progress impossible.
>
> A major difference between the AntMaze domains and other domains (HalfCheetah and Hopper) is that the AntMaze domains have sparse reward. In AntMaze, the agent is only rewarded with a positive reward if the agent successfully navigates to the goal location. In contrast, the HalfCheetah and Hopper domains have dense rewards and therefore even a sub-optimal policy can achieve decent performance.
>
> That said, several other algorithms such as BEAR [1] and EDAC [2] also attain poor performance in the AntMaze domains while performance in the HalfCheetah and Hopper domains are decent, making our results less surprising. We obtained their performance in the AntMaze domains and added them to Table 1. The performance on the HalfCheetah and Hopper domains are reported in the respective papers.
>
> [1] BEAR: https://arxiv.org/abs/1906.00949
>
> [2] EDAC: https://arxiv.org/abs/2110.01548
>
> ___
>
> **The experimental setting of the adding noise and bias to the AntMaze seems artificial, which is unfortunate because it underlies the primary result of the paper**
>
> We agree with the reviewer that testing the algorithms on more complex tasks would have made the case for our paper stronger. Nevertheless, we believe that our current experimental setting has the essence of a class of hard problems that require combination and stitching of various sub-optimal experiences. The AntMaze setting allows us to investigate the performance of offline RL algorithms in scenarios where support constraints are important, without other confounding challenges, such as learning from pixel input.
>
> As demonstrated in Table 1, current algorithms struggle with this setting, whereas our method can learn high-performing behaviors.

---

> > ### Author Response · Authors · 2022-08-04
> > **Dear Reviewer,**
> >
> > When you get the chance, can you please kindly have a look at our reply and let us know if your concerns have been addressed, or if you would like additional follow-up. Thank you!

---

> > > ### Author Response · Authors · 2022-08-07
> > > **Dear Reviewer**
> > >
> > > Can you please take a look at our reply and let us know if you still have concerns with the paper when you get the chance?
> > >
> > > There is only 3 days left in the author - reviewer discussion period, and we would love to hear if you are happy with our reply to your comments.
> > >
> > > Thank you!

---

> > > > ### Comment · Reviewer_Y2tj · 2022-08-07
> > > > **Reply**
> > > >
> > > > Appreciate the clarifications by the authors and the additional results provided. The artificial setting for the noisy AntMaze is the biggest concern I have, so I'll be sticking to my recommendation and rating.

---

> > > > > ### Author Response · Authors · 2022-08-07
> > > > > **Can you please have a look at Table 2?**
> > > > >
> > > > > **Results on AntMaze tasks when learning from standard datasets from D4RL in Table 2** demonstrates that our algorithm outperforms IQL by 15% and CQL by 18%.
> > > > >
> > > > > The AntMaze datasets used in Table 2 are used in numerous offline RL comparisons, and while the improvement (in terms of average overall score) on these tasks is not huge, the increment over prior works is well in line with improvements attained in other published papers (see, e.g., IQL, CQL versus Onestep RL, TD3+BC).
> > > > >
> > > > > Considering that we have provided additional results, would the reviewer by any chance consider increasing your score given the results in Table 2?
> > > > >
> > > > > Thank you very much!

---

### Meta-Review · Area_Chair_3gDW · 2022-08-22

**Recommendation:** Accept
**Confidence:** Certain

**Metareview:**

This paper tackles the important problem of learning policies that remain close to the data in offline RL. More specifically, the authors consider GAN-based methods. They provide a theoretical insight as to why prior work in modelling with GANs fail. Based on that, they are able to propose simple yet theoretically convincing modifications to GANs to train them properly. They conduct a thorough ablation study with evidence of the gain in performance brought by their proposed approach. Most questions were answered and clarifications were provided as part of the rebuttal. The artificial setting for the noisy AntMaze seems to be the biggest concern that remains. Overall, the paper was appreciated by reviewers.

**Award:**

No

---

### Decision · Program_Chairs · 2022-09-14

Accept